# Minimal residual disease detection by next-generation sequencing of different immunoglobulin gene rearrangements in pediatric B-ALL

Haipin Chen [1,2], Miner Gu [1,2], Juan Liang[1], Hua Song[1], Jingying Zhang[1], Weiqun Xu[1], Fenying Zhao [1], Diying Shen[1], Heping Shen[1], Chan Liao [1], Yongmin Tang [1] ✉ & Xiaojun Xu [1] ✉

While the prognostic role of immunoglobulin heavy chain locus (IGH) rearrangement in minimal residual disease (MRD) in pediatric B-acute lymphoblastic leukemia (B-ALL) has been reported, the contribution of light chain loci (IGK/IGL) remains elusive. This study is to evaluate the prognosis of IGH and IGK/IGL rearrangement-based MRD detected by next-generation sequencing in B-ALL at the end of induction (EOI) and end of consolidation (EOC). IGK/IGL rearrangements identify 5.5% of patients without trackable IGH clones. Concordance rates for IGH and IGK/IGL are 79.9% (cutoff 0.01%) at EOI and 81.0% (cutoff 0.0001%) at EOC, respectively. Patients with NGS-MRD < 0.01% at EOI or <0.0001% at EOC present excellent outcome, with 3-year event-free survival rates higher than 95%. IGH-MRD is prognostic at EOI/EOC, while IGK-MRD at EOI/EOC and IGL-MRD at EOI are not. At EOI, NGS identifies 26.2% of higher risk patients whose MRD < 0.01% by flow cytometry. However, analyzing IGK/IGL along with IGH fails to identify additional higher risk patients both at EOI and at EOC. In conclusion, IGH is crucial for MRD monitoring while IGK and IGL have relatively limited value.

Acute lymphoblastic leukemia (ALL) is the most common malignancy diagnosed in children. It involves chromosomal abnormalities and genetic alterations at an early stage of differentiation of lymphoid precursor cells[1]. Risk-adapted chemotherapy has improved the prognosis of childhood ALL remarkably for the past four decades[2–4]. By determining the patients' risk of relapse based on a reliable method, patients can be stratified and treated according to their risk. Minimal/measurable residual disease (MRD), defined as the presence of residual leukemic cells not detected by conventional tools, is a strong prognostic factor[5], and treatment strategies based on MRD status improve outcomes in childhood ALL[6]. There is compelling evidence associated with MRD negativity and the beneficial therapeutic outcome of long-term survival[7]. Treatment modalities can be tailored, and novel treatments may be administered based on MRD risk stratification to enhance the curative rate[8,9].

Multiparameter flow cytometry (MFC) or polymerase chain reaction (PCR) is widely applied for detecting MRD and essential in determining disease risks and directing therapeutic approaches for children with ALL[10,11]. MFC-based MRD measurements depend on the surface immunophenotype of the leukemia cells, and MRD measurements by PCR are based on the identification of leukemia-specific fusion genes/associated target genes. However, some leukemic clones

[1]Division/Center of Hematology-Oncology, Children's Hospital of Zhejiang University School of Medicine, The Pediatric Leukemia Diagnostic and Therapeutic Technology Research Center of Zhejiang Province, National Clinical Research Center for Child Health, No. 57 Zhugan Lane, Yan'an Street, 310003 Hangzhou, People's Republic of China. [2]These authors contributed equally: Haipin Chen, Miner Gu. ✉e-mail: y_m_tang@zju.edu.cn; xuxiaojun@zju.edu.cn

may have antigenic changes after therapy, and tumor heterogeneity may lead to false negativity, thereby limiting the application of MFC[12]. Recent studies applying next-generation sequencing (NGS) to monitor MRD have reported higher sensitivity and precision than MFC or quantitative PCR (qPCR) in hematological malignancies[13–16]. MRD monitoring using sensitive and robust NGS approach enhances the identification of relapse risk post-hematopoietic stem cell transplantation or chimeric antigen receptor modified T-cell therapy[17,18], and allows the identification of new emerging clones at each timepoint of the monitoring[19].

Immunoglobulin (Ig) gene rearrangements are a promising patient-specific target for the detection of MRD in B-lineage ALL (B-ALL) to identify the frequency and distribution of clonal sequences associated with a malignant lymphocyte population[4,15,20]. Previous clinical studies have mostly focused on the prognostic significance of NGS of the immunoglobulin heavy chain locus (IGH)[15,16,21], but there is a knowledge gap regarding the prognostic value of the immunoglobulin kappa (IGK) and lambda (IGL) light chain loci.

In this study, we aim to assess the prognostic value of both IGH and light-chain (IGK/IGL) MRD measurements in childhood B-ALL. To address this, we utilize a multiplex PCR and NGS-based technique to detect multiple B and T cell receptor sequences which encompass rearranged IGH (VDJ), IGH (DJ), IGK, IGKDE, and IGL. Herein, sequential detection of Ig gene rearrangements using NGS is conducted both at the time of diagnosis and during chemotherapy, and the prognostic values of IGH, IGK and IGL gene rearrangements are investigated.

## Results

### Baseline demographic and clinical characteristics

A total of 430 pediatric patients with B-ALL were enrolled between November 2018 and April 2022. As a part of the initial diagnostics, an Ig-/T-cell receptor (TCR) immune panel was analyzed to identify prognostic markers. Among these pediatric patients, 399 children (92.8%) had at least one trackable Ig clonal rearrangement and were included for further analyses. This cohort included 221 male and 178 female patients, with a median age of 4.4 (range: 0.3–17.8) years. The baseline characteristics are presented in Table 1. The estimated 3-year EFS and OS of 399 children in this cohort were 93.7% ± 1.6% and 96.7% ± 1.1%, respectively.

### Data quality evaluation

To statistically assess the data quality and stability across different time points and patient samples, the data for (1) number of input cells for sequencing, (2) size of sequencing raw data, (3) Q30 value of the FASTQ data, (4) IGK counts, and (5) IGL counts were subjected to data quality assessment using a 3-sigma (3-σ) test principle (Supplementary Tables 1 and 2 and Supplementary Fig. 1). The results confirmed the stability and reliability of the data. The IGK and IGL data from different patients at both diagnosis and follow up time points demonstrate good stability as well.

### Distribution pattern of Ig clonal rearrangement

Among 399 patients with B-ALL, 724 IGH clonal (including IGH-DJ) rearrangements in 377 children, 266 IGK clonal (including IGKDE) rearrangements in 176 children, and 83 IGL clonal rearrangements in 68 children were detected. Clonal rearrangements of ≥2 were found in 70.5% of children. The distribution of clones is shown in Fig. 1. Majority of the children displayed 1 to 3 clonal rearrangements (Fig. 1A), with Ig polyclonal pattern. Of the 377 children with trackable IGH clones, 42.7% and 34.2% presented with one and two IGH clones, respectively. Whereas among 176 and 68 children with trackable IGK and IGL clones, 67.0% and 80.9% presented only one clone, respectively (P < 0.001) (Fig. 1B). Among 399 children with trackable Ig clones, 188 displayed only IGH clones, 124 displayed both IGH and IGK clones, 34 displayed IGH and IGL clones, and 31 displayed IGH, IGK, and IGL clones (Fig. 1C). Twenty-two patients did not have trackable IGH clone but had IGL and/or IGK for tracking. Thus, the IGK/IGL rearrangements allowed tracking an additional 5.5% of samples that would have been uninformative if only the IGH assay was used for MRD monitoring. The distribution of clonotype numbers varied among different age groups (Fig. 1D). Specifically, in the IGH category, single clone rearrangement constituted the majority in patients over 10 years old, as opposed to those aged 1–10 years (67.3% vs 38.7%, P = 0.001). In contrast, of the IGK and IGL categories, majority of patients aged 1-10 years exhibited a dominance of one clone rearrangement which was comparable with those aged over 10 years old (70.5% vs 57.6% and 83.7% vs 72.2%, respectively, P = 0.095 and P = 0.237).

### Clearance patterns of Ig clones

The clearance of Ig clones was assessed at sequential timepoints of chemotherapy. As shown in Fig. 2, clonal clearance was observed in

**Table 1 | Demographic and clinical characteristics of patients**

| Characteristics | Number | Clonotypes | | | P value[b] |
|---|---|---|---|---|---|
| | | IGH (%) | IGK (%) | IGL (%) | |
| IG rearrangement | 399[a] | 377 (94.5) | 176 (44.1) | 68 (17.0) | |
| Gender | | | | | 0.722 |
| Female | 178 | 169 (94.9) | 76 (42.7) | 27 (15.2) | |
| Male | 221 | 208 (94.1) | 100 (45.2) | 41 (18.6) | |
| Age groups | | | | | 0.170 |
| <1 year | 7 | 7 (100.0) | 4 (57.1) | 1 (14.3) | |
| 1–10 years | 333 | 315 (94.6) | 139 (41.7) | 49 (14.7) | |
| ≥10 years | 59 | 55 (93.2) | 33 (55.9) | 18 (30.5) | |
| Gene fusions | | | | | 0.422 |
| ETV6-RUNX1 | 95 | 88 (92.6) | 47 (49.5) | 14 (14.7) | |
| TCF3-PBX1 | 25 | 25 (100.0) | 3 (12.0) | 3 (12.0) | |
| BCR-ABL | 17 | 15 (88.2) | 8 (47.1) | 4 (23.5) | |
| MLL rearrangement | 6 | 5 (83.3) | 2 (33.3) | 1 (16.7) | |
| Chromosome karyotypes | | | | | 0.295 |
| Normal | 146 | 139 (95.2) | 67 (45.9) | 30 (20.5) | |
| Hyperdiploidy | 125 | 118 (94.4) | 47 (37.6) | 15 (12.0) | |
| Others | 128 | 120 (93.8) | 62 (48.4) | 23 (18.0) | |
| Risk groups | | | | | 0.717 |
| Standard risk | 171 | 161 (94.2) | 73 (42.7) | 24 (14.0) | |
| Intermediate risk | 156 | 149 (95.5) | 66 (42.3) | 29 (18.6) | |
| High risk | 72 | 67 (93.1) | 37 (51.4) | 15 (20.8) | |
| WBC count | | | | | 0.300 |
| <100 × 10^9/L | 374 | 355 (94.9) | 165 (44.1) | 67 (17.9) | |
| ≥100 × 10^9/L | 25 | 22 (88.0) | 11 (44.0) | 1 (4.0) | |
| Stages | | | | | 0.306 |
| Progenitor B | 12 | 11 (91.7) | 4 (33.3) | 4 (33.3) | |
| Common B | 364 | 343 (94.2) | 166 (45.6) | 60 (16.5) | |
| Precursor B | 9 | 9 (100.0) | 1 (11.1) | 2 (22.2) | |

[a]Numbers in each category may not be total 399 due to missing data.
[b]P values obtained from comparing the composition ratios of all subgroups within this category using the chi-square or Fisher's exact test (two-sided).

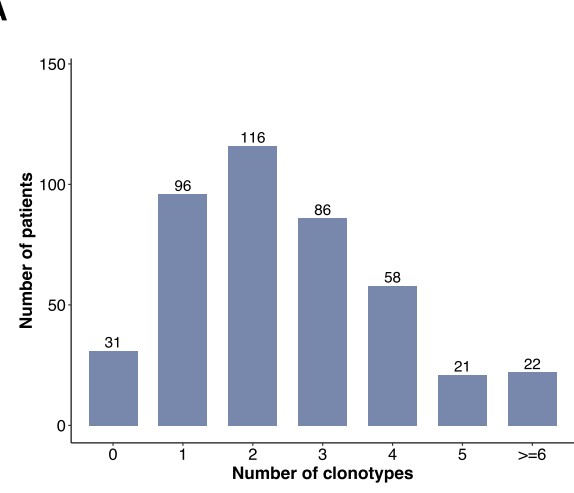

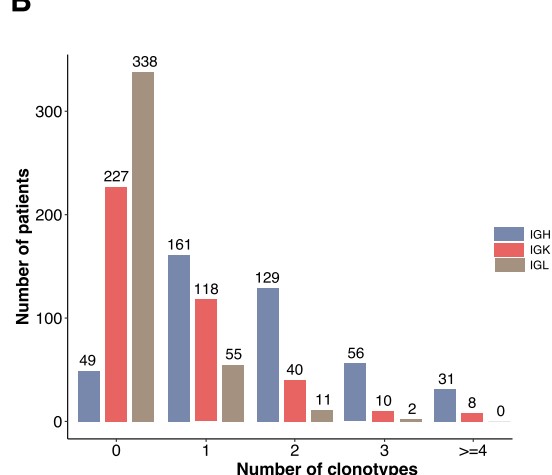

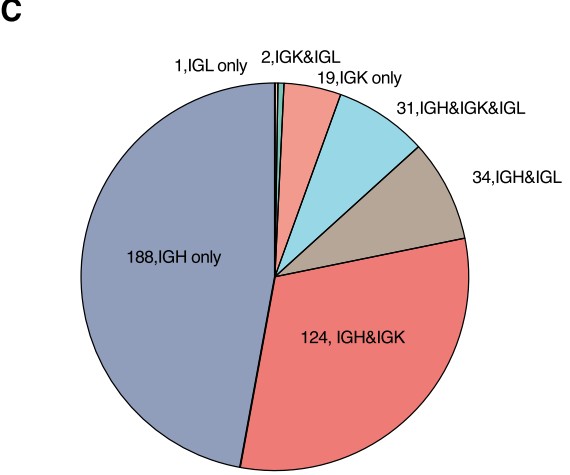

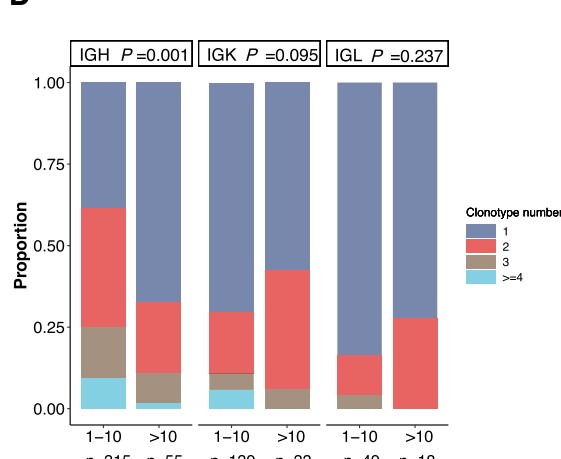

**Fig. 1 | Distribution characteristics of Ig clonal rearrangements. A** Distribution of total Ig clone numbers at the initial diagnosis. **B** Distribution of clone numbers of IGH, IGK and IGL. **C** The numbers and proportions of different combinations of clone types. **D** Distribution of clone numbers of IGH, IGK and IGL among different age groups (1–10 years (IGH $n = 315$, IGK $n = 139$, IGL $n = 49$) and >10 years (IGH $n = 55$, IGK $n = 33$, IGL $n = 18$)). Source data are provided as a Source data file.

60.9% of total clones at EOI, and another 31.4% of clone clearance was observed at EOC (Fig. 2A). Among the children who were clone negative at EOI, 1.83% (17/928) of clones became positive again at EOC. IGH, IGK, and IGL showed a similar pattern of clearance. Approximately 60% of the clones turned negative at EOI, and 21.6% to 32.6% of the clones continued to turn negative at EOC (Fig. 2B–D). The clone clearance time among IGH, IGK, and IGL did not exhibit a significant difference (Fig. 2E).

We then investigated the clearance patterns of clones with different sizes. The clearance rate of IGH clones with frequencies <10% was significantly faster compared with the clearance rate of clones with frequencies ≥50%, and the MRD clearance rate was higher at EOI (77.5% vs 54.2%; $P < 0.001$, Fig. 2F, G). At EOC, the proportion of children with positive IGH clones was lower but similar between the two groups (5.3% vs 6.7%; $P = 0.624$; Fig. 2F, G). The early clearance rates of IGK with frequencies <10% was also significantly faster than those of clones with frequencies ≥50% at EOI (69.4% versus 43.9%; $P = 0.019$; Fig. 2H, I). As for IGL, owing to the small sample size, the clearance rates of clones with frequencies ≥50% and frequencies <10% clones seemed to be comparable (80.0% versus 44.4%; $P = 0.170$; Fig. 2J, K). Similarly, when the exact clearance time were compared, IGH and IGK clones with frequencies ≥50% presented much lower clearance speed than those clones with frequencies <10% (Fig. 2L).

## Comparison of IGH-MRD and IGK/IGL-MRD

The clearance of IGH-MRD was compared with IGK/IGL-MRD. At EOI, if 0.01% was considered to be the cutoff values, the concordance rate of IGH-MRD and IGK/IGL-MRD was 79.9% (151/189; $P < 0.001$). At a higher sensitivity level (0.0001%), IGH-MRD and IGK/IGL-MRD had a concordance rate of 68.9% (130/189; $P < 0.001$) (Fig. 3A). At EOC, the vast majority of IGH, IGK, and IGL clones were completely eradicated or lower than 0.0001%. The concordance rate of IGH and IGK/IGL was 81.0% (153/189; $P < 0.001$) at the level of 0.0001% (Fig. 3B).

## Prognostic analysis of NGS-MRD

The prognosis of NGS-MRD was assessed at EOI and EOC. If any clonotype of IGH, IGK, or IGL was positive, NGS-MRD was defined as positive. We first investigated the associations between clinical/laboratory features and NGS-MRD eradication. As shown in Fig. 4A, B, patients aged older than 10 years or assigned into higher-risk group presented higher NGS-MRD levels at EOI, and patients in the high-risk group still had higher NGS-MRD level at EOC. Regarding the genetic features, patients with different fusion genes seemed to respond differently to induction therapy, but the multiple comparisons between different fusion genes did not reach statistical significance at EOI. (Fig. 4C). However, patients with *MLL*-rearrangements had higher NGS-MRD levels compared to other fusion genes, including *ETV6/RUNX1*,

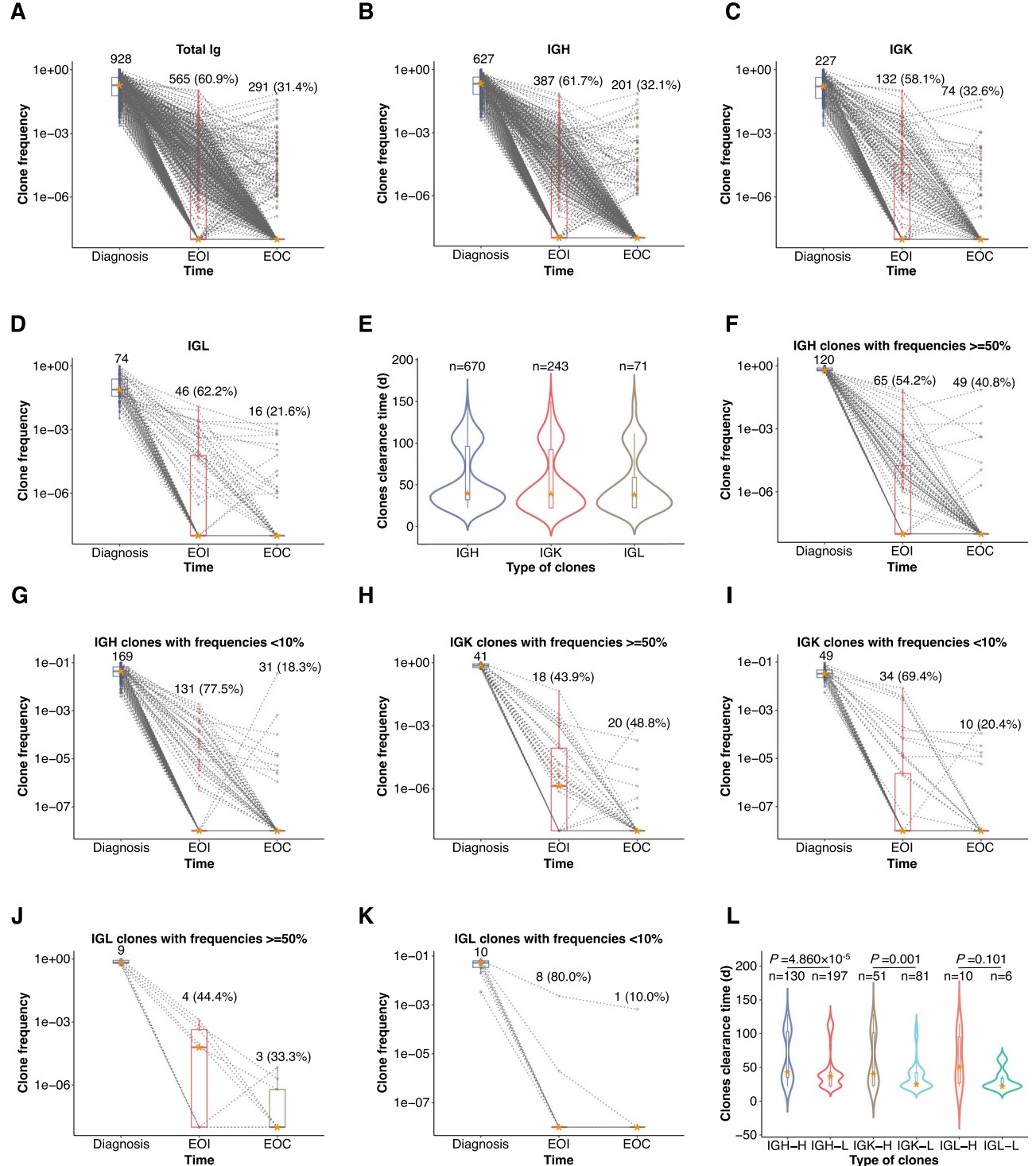

**Fig. 2 | Clonal clearance patterns of IGH, IGK and IGL and impact of clone frequencies.** The clearance patterns of total Ig clones and IGH, IGK and IGL are shown in (**A**–**D**). The numbers at diagnosis indicate the initial total clone numbers, and the numbers and ratios at end of induction (EOI) and end of consolidation (EOC) timepoints indicate clones turned to be undetectable. Only patients underwent NGS-MRD tests at all the three timepoints were included. Dotted lines connect the clones of the same patient at different timepoints. The box plots represent the interquartile range (IQR) of the data. The horizontal line and asterisk within the box indicate the median value. The vertical lines extending from the box, known as whiskers, represent the 1.5 times the IQR. Any data points outside this range are considered outliers. Violin plots show the clearance time of IGH ($n = 670$), IGK ($n = 243$) and IGK ($n = 71$) (**E**). The clearance time data were collected from the interim of induction, EOI, EOC. Timepoints after 150 days from the initial of treatment were excluded. The clearance patterns of IGH (**F**, **G**), IGK (**H**, **I**) and IGL (**J**, **K**) clones with frequencies ≥50% and <10% were compared as well. The total clone numbers at diagnosis, the clone numbers and ratios turned to be undetectable on EOI and EOC were marked. The clearance times of clones with frequencies ≥50% (IGH-H ($n = 130$), IGK-H ($n = 51$) and IGL-H ($n = 10$)) and those with frequencies <10% were compared (IGH-L ($n = 197$), IGK-L ($n = 81$) and IGL-L ($n = 6$)) using Wilcoxon signed-rank test (**L**). Source data are provided as a Source data file.

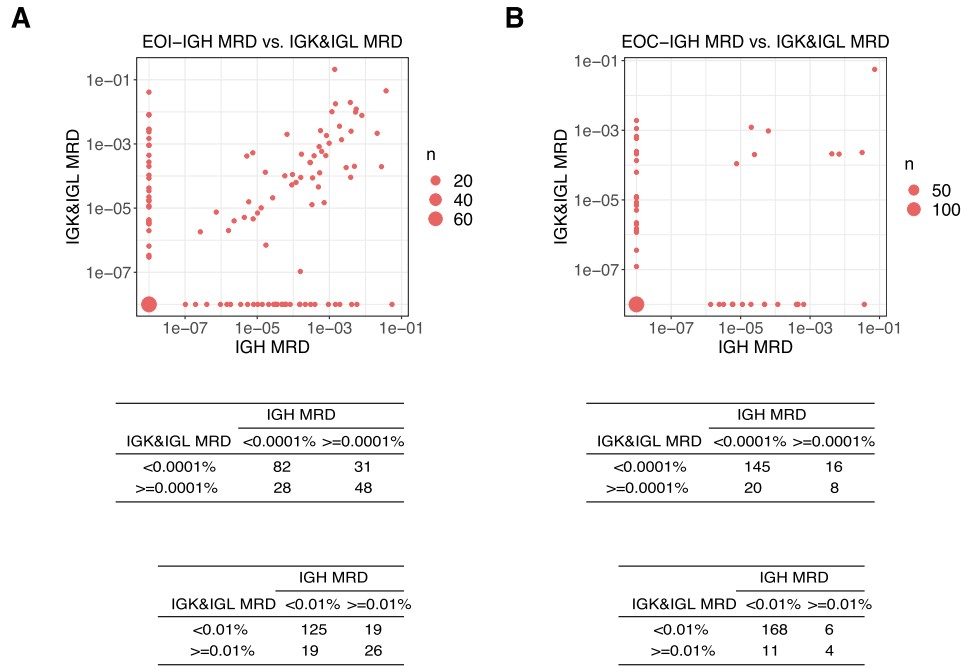

**Fig. 3 | Concordance between IGH-MRD and IGK/IGL-MRD.** The concordance between IGH-MRD (x-axis) and IGK/IGL-MRD (y-axis) at the EOI (n = 189) (**A**) and EOC (n = 189) (**B**) timepoints were assessed, and the corresponding numbers were calculated using cutoffs of 0.0001% and 0.01%, respectively. Source data are provided as a Source data file.

BCR/ABL1, and TCF3/PBX1 at EOC (Fig. 4D). The chromosome karyotype seemed not to be associated with NGS-MRD level at both EOI and EOC (Fig. 4C, D).

Regarding long-term survival, the 3-year EFS was 97.3% ± 1.3%, 96.0% ± 3.0% and 83.3% ± 6.2% (P = 0.002) in patients whose NGS-MRD were <0.0001%, ≥0.0001% and <0.01%, and ≥0.01% at EOI, respectively (Fig. 4E). The corresponding 3-year EFS was 96.4% ± 1.3%, 75.4% ± 14.4% and 83.4% ± 7.8% at EOC (P < 0.001; Fig. 4F), respectively. Multiple comparisons were performed between different MRD level groups, which suggested that 0.01% was the appropriate cutoff value for MRD at EOI, and 0.0001% MRD was a relevant cutoff value at EOC.

### Discordant MFC and NGS MRD and clinical implications
Next, we compared NGS with MFC for MRD detection. For patients with NGS-MRD < 0.01% at EOI, the 3-year EFS was 97.0% ± 1.2%, while those with MRD ≥0.01% was 88.2% ± 3.8%. In this cohort, MFC-MRD was negative (<0.01%) in 95.2% of patients at EOI (gating strategy for MRD test is shown in Supplementary Fig. 2). MFC-MRD was prognostic at EOI as well. Patients with negative and positive MFC-MRD presented 3-year EFS of 94.4% ± 1.8% and 78.9% ± 11.4% (P = 0.008), respectively (Fig. 5A). However, NGS identified 93 patients whose MFC-MRD was negative while their NGS-MRD was ≥0.01%, which accounted for 26.2% of all patients with negative MFC-MRD. These patients presented much worse 3-year EFS than those with both MFC and NGS MRD negative (84.7% ± 7.1% vs. 97.0% ± 1.2%, P = 0.008) (Fig. 5B). In contrast, there were only three patients whose NGS-MRD was negative but their MFC-MRD was ≥0.01%. All three patients survived at the last follow up. Patients with both MFC and NGS MRD positivity had the worst outcome, with 3-year EFS of 70.3 ± 15.4% only.

### Prognostic analysis of IGH, IGK and IGL MRD
When evaluating the MRD level solely based on the change of IGH alone, 351 children were included at EOI, and the 3-year EFS rates of children with IGH-MRD < 0.0001%, ≥0.0001% and <0.01%, and ≥0.01% were 94.7% ± 2.5%, 95.0% ± 3.7%, and 88.0% ± 4.4%, respectively (P = 0.034; Fig. 6A). In 377 children evaluated at EOC, the 3-year EFS

were 95.3% ± 1.9%, 86.1% ± 7.5%, and 74.0% ± 11.3% for those three groups, respectively (P < 0.001; Fig. 6B). The results implied that IGH-MRD had good prognostic significance. In 176 children assessed for IGK-MRD, the 3-year EFS was comparable among patients with IGK levels <0.0001%, ≥0.0001% and <0.01%, and ≥0.01% at EOI (95.2% ± 2.8% vs. 100.0% ± 0% vs. 84.6% ± 8.3%, P = 0.081, Fig. 6C) and EOC (93.4% ± 2.7% vs. 100.0% ± 0% vs. 100.0% ± 0%, P = 0.740, Fig. 6D). In 68 children assessed for IGL-MRD, the results indicated that IGL level was not prognostic at EOI (100.0% ± 0% vs.100.0% ± 0% vs. 77.8% ± 15.2% for IGL levels<0.0001%, ≥0.0001% and <0.01%, and ≥0.01%, P = 0.080, Fig. 6E), but was related to the 3-year EFS at EOC (100.0% ± 0% vs. 66.7% ± 27.2% vs. 83.3% ± 15.2% for IGL levels<0.0001%, ≥0.0001% and <0.01%, and ≥0.01%, P = 0.017, Fig. 6F).

### Evaluation on Integrating IGK/IGL into IGH MRD
In the literature, IGH rearrangement has been mostly used for NGS to evaluate the MRD level of B-ALL. The value of light-chain IGK and IGL has not been studied extensively, and their cumulative effects on MRD are not clear. Therefore, we first compared the differences in NGS and IGH MRD between EOI and EOC using a Sankey diagram. At EOI, by adding IGK/IGL results, 15 and 13 patients with IGH MRD <0.0001% were assigned to the NGS-MRD ≥0.0001% and <0.01%, and ≥0.01% groups, respectively, with 2 patients presenting events (Fig. 7A). At EOC, 13 and 7 patients whose IGH MRD <0.0001% were assigned to the NGS-MRD ≥0.0001% and <0.01%, and ≥0.01% groups, respectively, with 1 patient presenting event (Fig. 7B). Another 5 and 4 patients whose IGH MRD was between 0.0001% and 0.01% at EOI and EOC, respectively, were transferred to the NGS MRD positive ≥0.01% group according to their IGK/IGL results, without any relapse or death.

Next, we probed whether IGK/IGL tracking can further improve the accuracy of MRD by Ig NGS reports (Fig. 7C, D). We categorized the children with trackable IGH clones into 3 groups designated as (1) IGH positive (regardless of positive or negative IGK/IGL clones, IGH⁺), (2) IGH negative and IGK/IGL negative (IGH⁻IGK/IGL⁻), and (3) IGH negative but IGK/IGL positive (IGH⁻IGK/IGL⁺). Among the 351 children at EOI, 271 had IGH <0.01% (IGH negative), 18 had IGH <0.01% but IGK/IGL

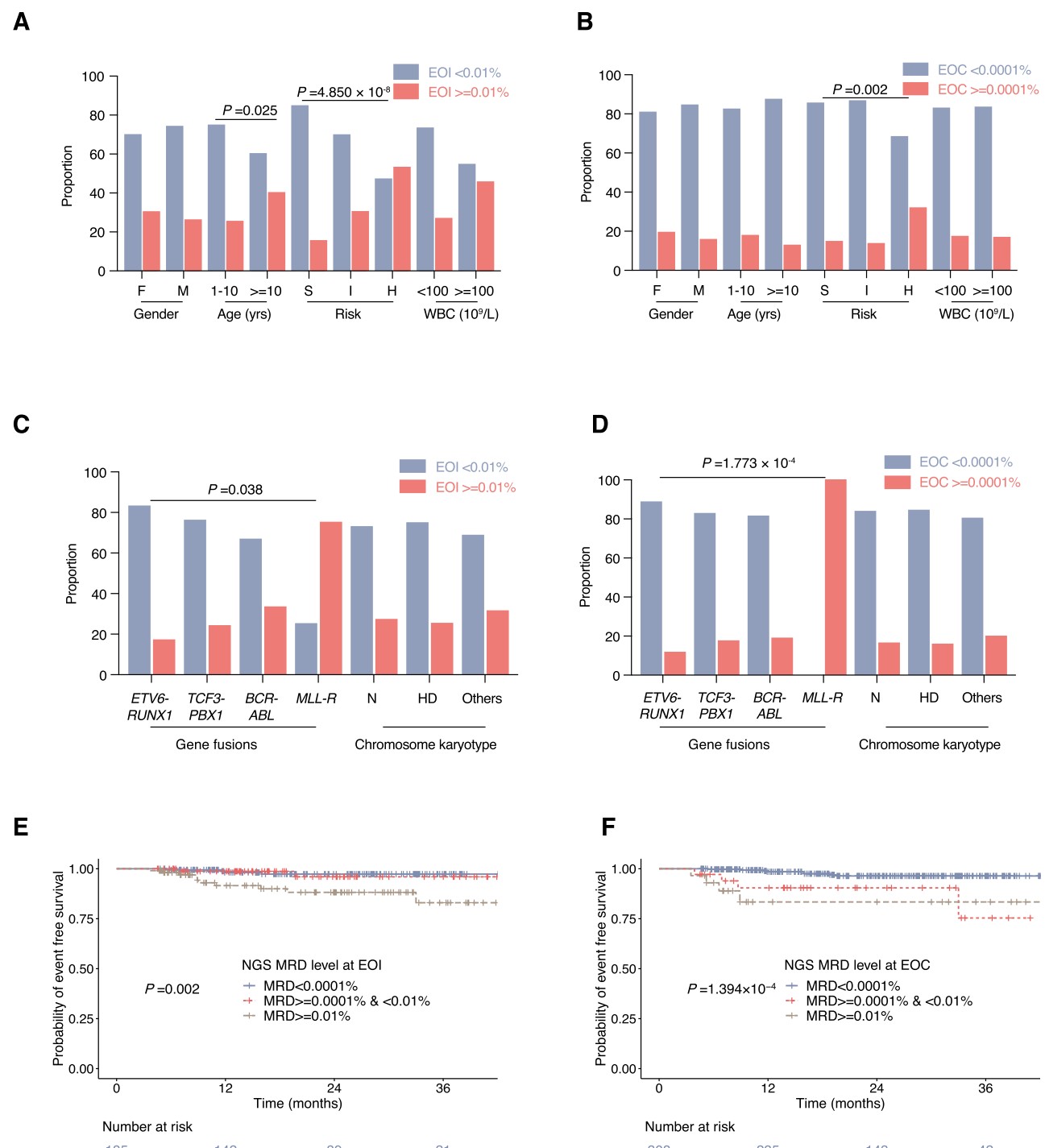

**Fig. 4 | Relationships between NGS-MRD and baseline clinical/laboratory features and survival.** Distributions of gender, age, risk groups, initial WBC count, fusion genes and chromosome karyotype between patients with NGS-MRD < 0.01% and ≥0.01% at EOI (**A**, **C**), and between patients with NGS-MRD < 0.0001% and ≥0.0001% at EOC (**B**, **D**) were compared. The total number of patients across various groups is as follows: At EOI, Gender 373, Age 367, Gene fusions 132, Chromosome karyotype 373, Risk groups 373, WBC count 373; At EOC, Gender 372, Age 365, Gene fusions 131, Chromosome karyotype 372, Risk groups 372, WBC count 372. The *P* values (two-sided) in (**A**–**D**) were obtained from comparing the composition ratios of all subgroups within this category using the chi-square or Fisher's exact test. Except for the *P* value marked, there is no statistically significant difference in the other groups. The Kaplan–Meier estimates of the event-free survival (EFS) at analytical cutoffs of 0.01% and 0.0001% for MRD as measured by NGS at the end of induction (EOI) (*n* = 373) (**E**) and the end of consolidation (EOC) (*n* = 372) (**F**). *P* values in **E**, **F** were calculated using two-sided log-rank test. F female, M male, S standard risk, I intermediate risk, H high risk, WBC white blood cells, *MLL*-R *MLL* rearrangement, N normal, HD hyperdiploidy. Source data are provided as a Source data file.

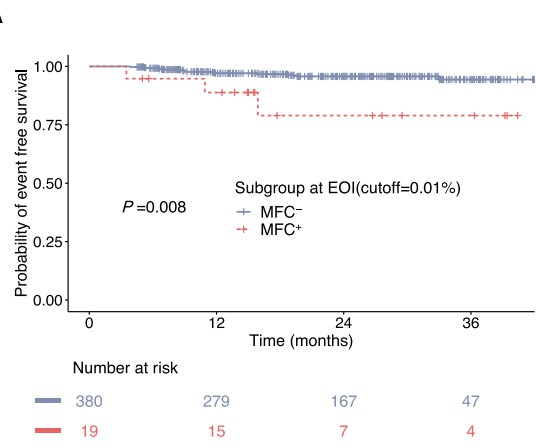

**Fig. 5 | Discordant MFC and NGS MRD and clinical implications. A** Kaplan−Meier estimates of event-free survival (EFS) at analytical cutoffs of 0.01% for MRD as measured by multiparameter flow cytometry (MFC) at the end of induction (EOI) (*n* = 399). *P* value was calculated using stratified log-rank test. **B** Kaplan−Meier estimates of EFS for three MRD groups (NGS-MFC⁻, NGS⁺MFC⁻ and NGS⁺MFC⁺) based on NGS and MFC MRD at EOI (*n* = 369), using analytical cutoffs of 0.01% for MRD, which identified a cohort of 93 patients who were negative for MRD by MFC

but positive by NGS presenting an intermediate prognosis. Three patients presenting NGS⁻MFC⁺ MRD were not included for comparison, and all of them underwent event free survival at last follow up. *P* value was calculated by the two-sided log-rank test. Bonferroni adjustment was performed for multiple comparison, with *P* < 0.0167 considered statistically significant. Source data are provided as a Source data file.

≥0.01% (IGH⁻IGK/IGL⁺), and two of the 18 patients relapsed. The EFS of children with IGH⁻IGK/IGL⁺ was comparable with that of those with IGH⁻IGK/IGL⁻ (73.3% ± 17.6% vs 96.8% ± 1.3%, *P* = 0.032, not statistically significant after Bonferroni adjustment). At EOC, when 0.0001% of IGH, IGK, and IGL was taken as the cutoff value for MRD negativity, the 3 groups were also compared. The results showed that 20 of the 312 IGH-negative children were IGK/IGL positive, and one child relapsed. The EFS of IGH⁻IGK/IGL⁺ children was comparable to that of IGH⁻IGK/IGL⁻ children as well (75.0% ± 21.7% vs 96.6% ± 1.3%; *P* = 0.327).

## Discussion

Monitoring minimal residual disease (MRD) is a standardized and well-accepted method for measuring disease status in patients with ALL, and it has become an integral part of diagnostic patient care[22,23]. In addition to serving as an independent prognostic factor, MRD helps guide risk stratification and treatment decisions based on its dynamic change[24,25]. However, most conventional MRD techniques are insufficiently standardized or sensitive in detecting low levels of MRD. While MFC and qPCR are the most commonly used techniques, they also have limitations. MFC is readily available and fast, but due to the heterogeneity of ALL, it requires cell surface markers that may not be sensitive or specific enough[26]. On the other hand, while qPCR of Ig/TCR offers high sensitivity and specificity, it is time-consuming and has limitations related to the choice of primers and changes in clones, which can lead to false negative results[27]. In contrast, NGS with its high-throughput sequencing and ultra-sensitivity overcomes these limitations, with sensitivity levels that can reach up to $10^{-6}$. This high sensitivity allows for the identification of patients in the very-low-risk group and expands the intervention window for those at risk of relapse[14].

Ig and TCR gene rearrangements are clone-specific molecular markers used in MRD analysis. The use of oligoclonal Ig/TCR targets in MRD analysis may result in underestimation of the ALL load due to continuous and secondary rearrangements[28,29]. Thus, two MRD markers with high sensitivity were recommended to reduce the false-negative results arising from clonal evolution in previous studies around or before the year 2000[20]. In contrast, NGS provides comprehensive analysis of all kinds of clonal rearrangements[30]. Whether the combination of IGH targets with other Ig rearrangements like IGK and IGL is a more suitable strategy for sensitive MRD detection in the era of NGS should be clarified.

Studies have shown that monitoring B-ALL patients at different timepoints using IGH, IGK (VK-KDE), TCRG, and TCRD gene rearrangements achieves acceptable sensitivity levels[31]. In this cohort of pediatric patients with B-ALL, it was observed that 7.4% of the cases did not have dominant Ig clonal rearrangement for MRD tracking, which was similar to a previous report in adult ALL[32]. The possible reasons for this could be attributed to the following two aspects: First, malignant cells transformation may occur prior to Ig gene rearrangements, rendering them undetectable through this method[15,33]. Secondly, mutations at the primer binding sites of the assay may result in ineffective and nonspecific amplification[34,35].

Although clinical studies have used NGS to monitor IGH rearrangement in B-ALL[15,16], studies on IGK and IGL are limited, and most previous studies have tracked IGK and IGL rearrangements by qPCR, resulting in controversial results[4]. In this study, we investigated the distribution pattern of IGK and IGL and assessed whether the monitoring of IGH along with IGK and IGL had any complementary significance. Our cohort showed that IGH, IGK, and IGL clonal rearrangements were found in 94.5%, 44.1%, and 17.0% of children, respectively, which was similar to previous reports[4,10]. However, we found that IGK and IGL were not independent factors to monitor MRD, and IGH was the only marker with strong prognostic significance at both EOI and EOC, which was quite similar to another study conducted in adult ALL patients[32]. The cutoff value of its application was set at different sensitivities. MRD less than 0.01% was the most important indicator for excellent outcome, while it would be beneficial if MRD was negative or less than 0.0001% at EOC. Unfortunately, at EOI, we did not find any correlation between IGK or IGL clearance and EFS in children with residual IGK and IGL. As IGK and IGL can only cover a small population and their independent prognostic significance is not clearly noticeable, they are not suitable for the evaluation of MRD independently.

As IGK and IGL alone were not sufficient indicators of MRD, whether they can be combined with IGH to further improve the evaluation efficiency of IGH was investigated. In our study, among the children who were negative for IGH at EOI, 6.6% (18/271) and 7.1% (22/312) were positive for the light-chain clones IGK and IGL, respectively. Patients with IGH⁻IGK/IGL⁺ MRD presented similar EFS to those with IGH⁻IGK/IGL⁻ MRD both at EOI and EOC. This indicates that the detection of IGK and IGL presented limited value to IGH monitoring. Unlike

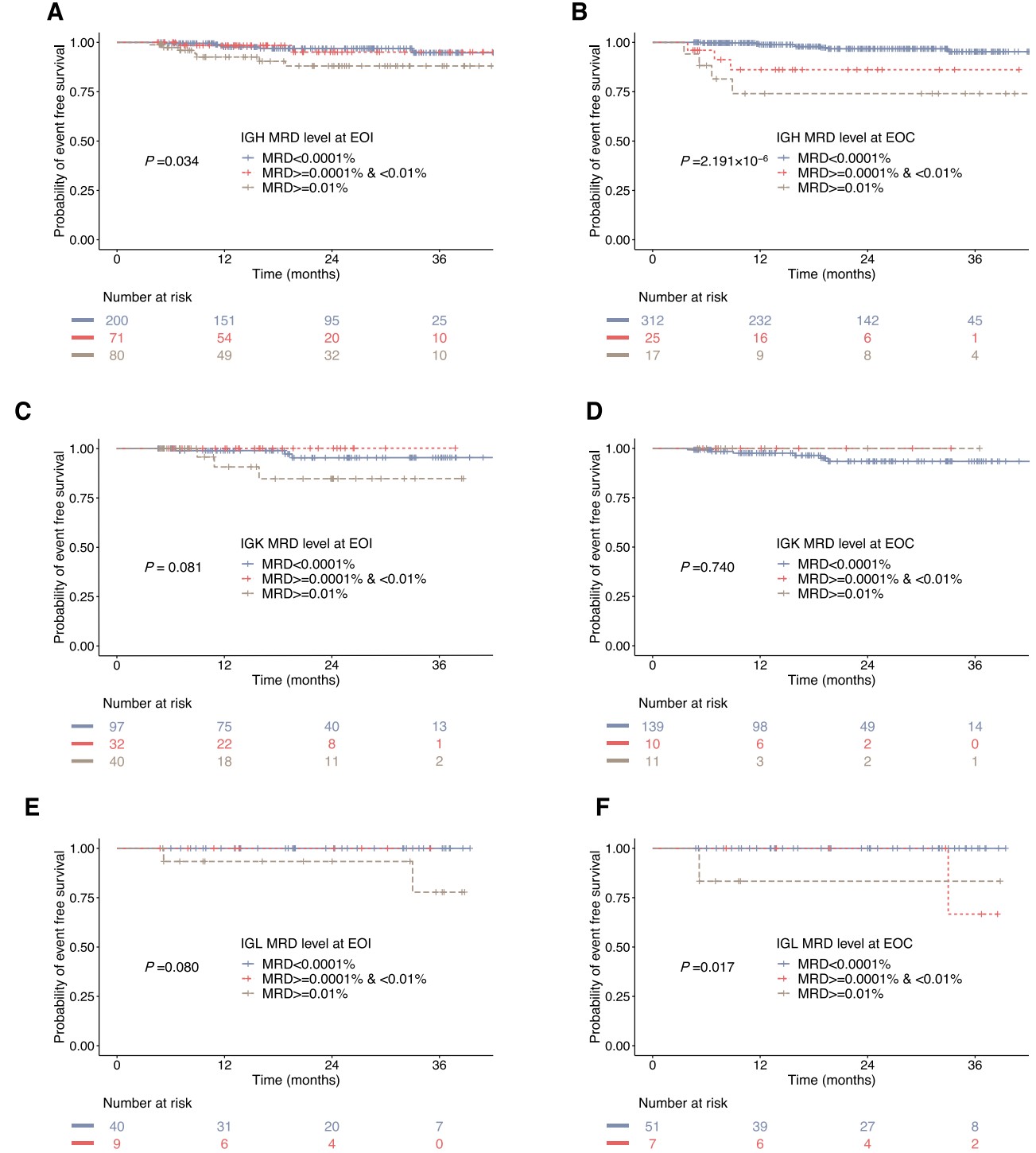

**Fig. 6 | Prognostic values of IGH, IG K and IGL MRD.** Kaplan−Meier estimates of the event-free survival (EFS) at analytical cutoffs of 0.01% and 0.0001% for IGH-MRD as measured by NGS at the end of induction (EOI) ($n = 351$) (**A**) and the end of consolidation (EOC) ($n = 354$) (**B**). Kaplan−Meier estimates of the EFS at analytical cutoffs of 0.01% and 0.0001% for IGK-MRD as measured by NGS at EOI ($n = 169$) (**C**) and EOC ($n = 160$) (**D**). Kaplan−Meier estimates of EFS at analytical cutoffs of 0.01% and 0.0001% for IGL-MRD as measured by NGS at EOI ($n = 65$) (**E**) and EOC ($n = 64$) (**F**). *P*-value was calculated using log-rank test. Source data are provided as a Source data file.

NGS, which could identify 26.2% of patients with intermediate-risk from those with negative MFC-MRD, analyzing IGK and IGL along with IGH identified only another 7% of patients with positive light-chain but similar outcome to IGH⁻IGK/IGL⁻ patients. Thus, we considered that integrating IGK and IGL levels into the IGH MRD added relatively limited value to the IGH-based risk stratification system.

Previous studies on B-ALL have generally evaluated the prognostic significance of NGS-MRD at EOI timepoints but have not focused much on EOC[15,16]. While the EOI-MRD measurement is important for prognosis and risk stratification, assessing MRD at later timepoints can add value for clinical management[36]. In our study, children with MRD levels <0.01% at EOI had a good prognosis. However, at the EOC, the

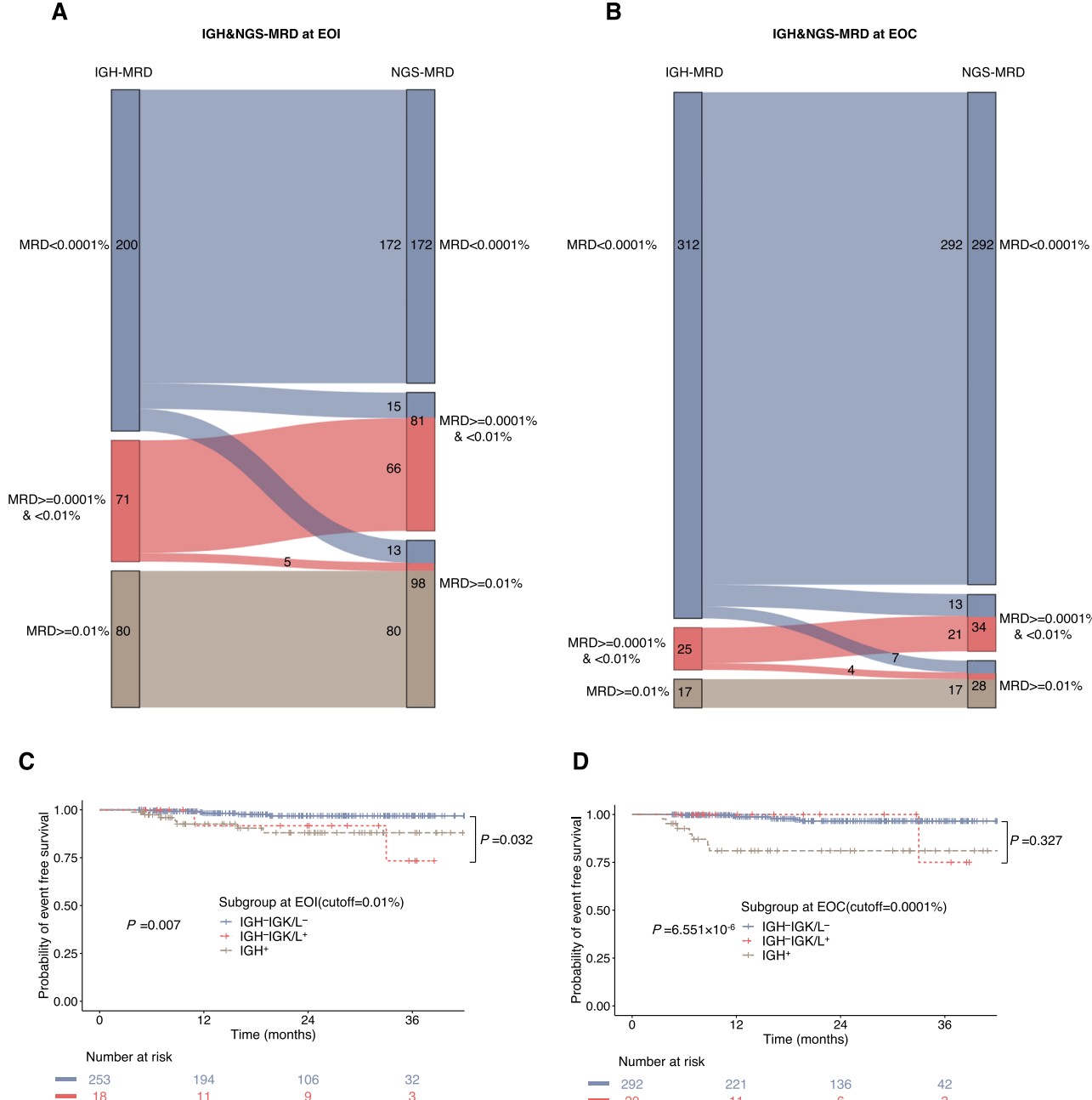

**Fig. 7 | Evaluation on Integrating IGK/IGL into IGH MRD.** The discrepancy in IGH-MRD and NGS-MRD (NGS-MRD is the sum of IGH, IGK and IGL MRD) at the end of induction (EOI) ($n$ = 351) (**A**) and the end of consolidation (EOC) ($n$ = 354) (**B**) shown by Sankey diagram. Due to the discordance of IGH and IGK/IGL MRD levels, patients were divided into three groups according to their IGH and IGK/L levels: IGH$^+$, IGH$^-$IGK/IGL$^-$, and IGH$^-$IGK/IGL$^+$. Kaplan–Meier estimates of the event-free survival (EFS) for the 3 groups at EOI ($n$ = 351) (**C**) and EOC ($n$ = 354) (**D**). Bonferroni adjustment was performed for multiple comparison, with $P$ < 0.0167 considered statistically significant. Source data are provided as a Source data file.

MRD detection limit must be <0.0001% to show a favorable outcome. At EOC, 82.3% of patients achieved MRD below 0.0001%. For the remaining 17.7% of children whose leukemia clones were not cleared, even if their MRD is below 0.01%, advanced therapeutic intervention such as adding immunotherapy modalities, such as blinatumomab, is necessary, given its superior efficacy in MRD eradication compared to traditional chemotherapy. In the adult ALL E1910 study, the addition of blinatumomab to consolidation therapy also showed good survival benefits for MRD <0.01% (detected by MFC) patients, suggesting that blinatumomab may further improve survival by deepening the depth of MRD remission[37]. In another clinical trial conducted in children with

high-risk or first-relapsed B-ALL, even patients with MRD less than 0.01% showed a trend of benefiting from blinatumomab when compared with traditional chemotherapy, indicating that deeper MRD must be eradicated to prevent relapse[9]. As our study established that NGS is both sensitive and high throughput, detecting deeper MRD remission by NGS is clinically significant in guiding the application of immunotherapy.

In addition to Seq-MRD testing method used in this study, there are other available MRD assays that capture the immunoglobulin heavy chain (IGH) and kappa/lambda light chain (K/L) loci, such as Adaptive Clonoseq and Invivoscribe. These three methods share a similar overall

objective of detecting and monitoring MRD in B-cell malignancies, and they all demonstrate high sensitivity reaching $10^{-6}$. Seq-MRD and Clonoseq target the IGH (VDJ), IGH(DJ), IGK, and IGL loci to identify clonotypes using sequencing data, while Invivoscribe focuses on the IGH and IGK loci. Moreover, the definition of the dominant clone for following tracking was the same between Seq-MRD and Clonoseq, which is different from that of Invivoscribe[38,39].

NGS for MRD assessment is applicable to various B-lineage malignancies except for ALL, such as chronic lymphocytic leukemia (CLL), multiple myeloma (MM) and lymphoma[16,35,40,41]. It plays a crucial role in improving risk stratification, treatment response monitoring, and prognostic outcomes[16,42]. NGS enables more sensitive and specific detection of MRD, allowing for precise risk assessment and identification of patients at higher risk of relapse[14,43]. It facilitates personalized treatment strategies, such as targeted therapies and transplantation, based on MRD status[41,44]. NGS-based MRD assessment provides valuable information for clinical decision-making and enhances the management of these diseases.

Our present study has certain limitations. First, the follow-up time of the cohort was relatively short. Although our result of the prognosis was consistent with previous research, the exact conclusion needs to be observed for a longer period. Second, the small sample size of patients with IGK and IGL limited our ability to draw definitive conclusions. Future studies with extended observation periods and larger population will provide a more comprehensive understanding of the prognostic significance of MRD monitoring in this patient population.

In conclusion, the results of our study indicate that NGS of Ig is an improved analytic platform for the measurement of post-treatment MRD in pediatric B-ALL. Patients with NGS-MRD <0.01% at EOI or <0.0001% at EOC present a nearly curable disease. With high availability and sensitivity, the clonal rearrangement of IGH is the most important for MRD monitoring, whereas the clonal rearrangement of IGK and IGL was found only in approximately half of the cohort with very limited independent prognostic significance. Thus, we recommend continuing to pursue IGH monitoring but not IGK/IGL monitoring according to our experience.

## Methods

### Study design, patients, and procedures

This study was approved by the Institutional Review Board of Children's Hospital, Zhejiang University School of Medicine and was conducted in accordance with the Declaration of Helsinki regulations and the International Conference on Harmonization and Good Clinical Practice guidelines (ClinicalTrials.gov Identifier: NCT05973032). Informed consent was obtained from the parents/guardians of each patient. No participant compensation was provided. Sex and/or gender of participants was not considered in the study design; more than half of participants were male (240 of 430). Sex of participants was determined based on physical examination and chromosome karyotype.

This was a single-center, observational study conducted in children with ALL between November 2018 and June 2022. Children ≤18 years with newly diagnosed B-ALL who undergone NGS of B-cell receptors at the Children's Hospital of Zhejiang University School of Medicine were included in this study. Patients with extramedullary leukemia at diagnosis were excluded. The European Group for the Immunological Characterization of Leukemias (EGIL) criteria were applied to diagnose and classify ALL in this study. All enrolled patients were treated according to the ZJCH-ALL-2019 protocol detailed in the Supplementary Fig. 3 and Table 3. This protocol was implemented in our center in September 2018 and subsequently extended to all of Zhejiang Province in 2019. In this protocol, NGS-MRD was not used for patient risk stratification or treatment allocation. For the detection of MRD, bone marrow (BM) aspiration for Ig NGS was collected at diagnosis, the end of induction (EOI) at the 5th week from the initial prednisone prephase, and the end of consolidation (EOC) at the 13th week which was before the start of early intensification for low and intermediate risk patients, and was before the start of the second course of consolidation for high-risk patients. NGS-MRD was sequentially monitored every 2–3 months after consolidation until it was undetectable. In this study, NGS-MRD refers to the quantitative value of MRD detected through NGS testing which was the sum of IGH and light chain (IGK/IGL) levels. NGS detection during other timepoints, such as the interim of induction and timepoints after NGS-MRD was negative, was not mandatory but monitored as per parents' preference. The patients were followed up until August 30, 2022 and the median follow-up time was 20.7 months.

### Detection of MRD by NGS

DNA was extracted from bone marrow samples and sent for sequencing using Seq-MRD® technology (ImmuQuad Biotech in Hangzhou, China). The details of the sequencing method are as follows. The input DNA for diagnostic samples (clonality test) was 500 ng, while for post-treatment samples (MRD tracking test), it was at least 7 µg, which corresponded to $1 \times 10^6$ mononuclear cells (MNCs). To amplify all possible rearranged sequences in the samples, a specific set of primers was used, including 23 primers for the IGH gene, 9 primers for the incomplete IGH gene, 15 primers for the immunoglobulin kappa locus (IGK), including KDE, and 17 primers for the immunoglobulin lambda locus (IGL). These primers targeted the V, D, and J regions of the IGH, IGK, and IGL genes, respectively. For sample identification, primers with sample identifiers were used in the second round of PCR amplification. Subsequently, the PCR products with identifiers were purified using AMPure XP beads (Beckman, USA) and combined into a sequencing library. The mixed library was then sequenced using the Illumina® Novaseq PE150 platform.

The Ig-Blast algorithm was used to determine Ig clonotypes, and the obtained CDR3 sequences were compared with the IMGT database. Clonotype frequencies were calculated by dividing the number of sequencing reads for each clonotype by the total number of passed sequencing reads within a sample. A frequency threshold of 3% was used to define dominant clone gene rearrangements. If a sequence accounted for at least 3% of all sequences at a given locus, represented at least 0.02% of total nucleated cells in the sample, significantly separated from the background clone repertoire, and had sufficient uniqueness for tracking, it was considered suitable for tracking[45].

For enhanced result accuracy, each testing batch included an internal standard. This standard comprised a positive control with known rearrangements spike-in DNA, which allowed assessment of PCR amplification efficiency. Furthermore, during the development of the NGS MRD methodology, comprehensive optimization and validation processes were carried out to ensure reliability and validity. To mitigate primer amplification bias, primers were optimized using over 200 spike-in sequences synthesized from plasmids. Adjustments were made to primers with low amplification efficiency by varying PCR enzymes, buffers, temperatures, primer melting temperatures (Tm), and reaction cycle numbers. Mixed known DNA samples served as positive controls during sample testing to assess batch-specific amplification performance.

The linearity of the NGS MRD method was assessed by generating samples with known absolute copy numbers and rearrangements. Cell line DNA samples at different concentrations were mixed with healthy individuals' PBMC DNA samples. The detected results were compared to the expected results to confirm the linear relationship.

The uniqueness of the dominant clone sequences was assessed by establishing a healthy donor Ig database. The probability of occurrence of the detected dominant clone in the healthy donor Ig database was investigated, and each dominant clone sequence was assigned a uniqueness index. This index provided an evaluation of the likelihood of the sequence being present in the non-ALL repertoire. Clones with low

uniqueness indices were excluded from MRD tracking. Different clone sequences in the tested samples have varying limit of detection (LoD). When determining the LoD for trackable sequences, several factors were taken into account, including the amount of DNA input, sequence uniqueness, as well as the diversity and mutation of those sequences.

### Statistical analysis

The association between categorical variables was tested using $\chi^2$ test, the correlation between quantitative variables was measured using Pearson correlation and tested using Student's $t$ distribution, and ANOVA was used to compare quantitative variables. Event-free survival (EFS) and overall survival (OS) curves were estimated using the Kaplan–Meier method and compared according to the log-rank test. Death during induction, abandonment before complete remission, death in continuous complete remission, relapse, and secondary malignancies were considered as events in the calculation of EFS probability. The EFS time was calculated from the date of diagnosis to the last date of follow-up or the first event. The OS was calculated from the date of diagnosis to death from any causes with censoring the patients alive at the time of data analysis. The final date for follow-up was August 30, 2022. Data visualization was performed in R version 4.0.3 using the following packages: ggplot2, survival, survminer and networkD3 package and GraphPad Prism 9.0.0. Statistical analysis was performed in Graphpad Prism 9.0.0 and SPSS software (version 23.0.0.0). A $P$ value < 0.05 (two tailed) was considered to be statistically significant. Bonferroni adjustment was performed for multiple comparisons.

### Reporting summary

Further information on research design is available in the Nature Portfolio Reporting Summary linked to this article.

## Data availability

The export of genetic information (raw sequencing data) and materials relevant to this work have been approved by the Ministry of Science and Technology of China. The raw sequencing data of this research are available under restricted access at the Genome Sequence Archive (GSA) for Human, Project ID HRA005729, and can be found at https://ngdc.cncb.ac.cn/gsa-human/browse/HRA005729. The reason for the restricted access to the data is to protect individual genetic information, to ensure that researchers comply with the corresponding regulations when using the data, and to prevent its use for commercial purposes. Each application for data access will be reviewed individually; access to the data is intended for scientific research purposes and is available predominantly to researchers affiliated with an accredited institution, who can demonstrate a legitimate scientific purpose. Applicants must submit a detailed research plan to the corresponding author, clarifying the objectives of their proposed study and affirming their commitment to non-commercial use of the data. They must also agree not to redistribute the data. Whether a request is approved will depend on the consistency of the proposed research with ethical guidelines and data usage agreements. Once access is granted, the data will be available to researchers for the duration agreed upon within the data access agreement, typically for no longer than one year, with the possibility of extension upon request. The commitment is to respond to access requests within 10 business days. We are committed to facilitating prompt access to the data after receiving complete and adequately justified requests. The remaining data pertinent to this study are included within the Article, Supplementary Information, or the Source data file and are freely accessible without restrictions. Source data are provided with this paper.

## Code availability

The data were analyzed in R version 4.0.3 using the following packages: ggplot2, survival, survminer and networkD3 package. The code used in this study has been deposited at https://github.com/pediatric-B-ALL-NGS/pediatric-B-ALL-NGS.

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

## Acknowledgements

This study was supported in part by grants from the Key Project from Science and Technology Department of Zhejiang Province (No. 2019C03032, (Y.T.)) and the Pediatric Leukemia Diagnostic and Therapeutic Technology Research Center of Zhejiang Province (No. JBZX-201904, (W.X.)).

## Author contributions

H.C., M.G., J.L., H. Song, J.Z., W.X., F.Z., D.S., H. Shen, C.L., and X.X. were the responsible pediatricians and participated in data collection; H.C., M.G., and X.X. performed data analysis; Y.T. and X.X. designed the work and performed data interpretation; H.C., M.G., and X.X. drafted the manuscript; Y.T. and X.X. revised the manuscript; and all authors reviewed the manuscript and provided final approval.

## Competing interests

The authors declare no competing interests.
