## [Peer Review File · Nature Communications]

Minimal residual disease detection by next-generation sequencing of different immunoglobulin gene rearrangements in pediatric B-ALLReviewers' Comments:

Reviewer #1:

Remarks to the Author:

The treatment protocol should have been in the supplementary material, but I was not able to find it. However, I understand from previous publications that B-ALL patients with MRD >1% at day 15 (TP1) or >0.1% at day 33 (TP2) of induction phase were allocated to the HR Group. Was NGS MRD used to stratify the patients of the cohort here reported? This aspect should be clarified in the manuscript. There is also an inconsistency on treatment; it is stated that patients in this cohort were treated according to the ZJCH-ALL-2019 protocol, while patients here reported were diagnosed since 2018. In respect of clinical outcomes and correlation with MRD findings it should be recognized that the follow-up time is rather short (median 20 months).

In the Discussion it is stated that in qPCR MRD monitoring "To reduce false-negative results arising from clonal evolution, two MRD markers with high sensitivity are generally required in clinical trials." While this approach was used in the 2000, it is no longer applied; one sensitive marker is generally considered adequate for patients' stratification.

Concerning the NGS MRD methodology used, there is no information about the presence of an internal standard, such as the spike-in of a DNA sample with known rearrangements of IGH/IGK/IGL and a known number of absolute copies of these rearrangements. The absence of such a type of spike-in, which instead is applied for example in the study by Svaton et al. (ref.16), may decrease the accuracy of results. Particularly at the EOI and EOC points, when the levels of disease are very low or negative, the amplification derived from MNC and residual blasts may not be linear, and this can limit the accuracy of MRD levels assessment. Thus, the authors should clarify if they used an internal calibrator and if not, whether they demonstrated the linearity of rearrangements amplification.

It should be also clarified if data on MRD by PCR or FLOW are available in these patients, and if yes, to compare those data with NGS data to establish how they correlate with NGS findings.

A novelty of this paper regards the findings on IGH vs. IGK/IGL, which suggest that IGK/IGL MRD findings are poorly correlated to prognosis, in favor of IGH. In the conclusion remarks the authors should indicate if in their opinion it is worth to continue pursuing IGK/IGL monitoring.

Editing suggestions:

Line 45, Background: "Monitoring the levels of minimal/measurable residual disease (MRD) defined as the presence of residual leukemic cells not detected by conventional tools, is a strong prognostic factor": "Monitoring" should be "Detection".

Line 93. "per the parent's preference" should be "per parents' preference"

Line 100: "The first-step PCR primers were aimed to acquire": there is a need for editing; maybe "The first step was aimed to acquire PCR primers for"

Line 127: clonotypes should be clones

Line 154: "As a part of the initial visit," should be "As a part of the initial diagnostics,"

Line 190: "The clearance of Ig clones was observed after chemotherapy." maybe better "The clearance of Ig clones was assessed at sequential timepoints of chemotherapy."

Line 191-192: both percentages should be referred to clones (and not to children)

Reviewer #2:

Remarks to the Author:

This is a large cohort of pediatric ALL patients evaluated with modern MRD assays. The design of the study is robust and the results are relevant. The paper is well written with balanced interpretations

and references. The authors address the limitation that follow-up time of the cohort was relatively short.

Minor comments:

1. In the manuscript, the authors state "...A total of 430 pediatric patients with B-ALL were enrolled between November 2018 and April 2022. As a part of the initial visit, an Ig-/T-cell receptor (TCR) immune panel was analyzed to identify prognostic markers. Among these pediatric patients, 399 children (92.6%) had at least one Ig clonal rearrangement and were included for further analyses...". What is the reason for 7.4% of the cases not having at least one Ig clonal rearrangement? Sample quality or issues with the assay, or both? This should be briefly commented/clarified in the paper; both in the Results and Discussion sections.

2. Would the authors be able to genomically profile cases with adverse (vs superior) clinical outcomes, e.g. using WGS? Similarly, can the authors profile host immune profiles in a similar manner? Such approaches would provide more detailed biological insights on biology in relation to outcomes. If possible, please add these data and expand the paper. If not possible, this should be commented briefly in the Discussion section.

3. How do the applied MRD assays perform in relation of other available MRD assays capturing IGH and K/L, such as Adaptive Clonoseq, Invivoscribe, etc? How are the assays similar/different? What data is available from ALL patients (or other B-cell malignancies) using these type of assay? For example, work has been done in lymphomas by MSKCC, Stanford, and NIH/NCI researchers, and work has been done in myeloma by MSKCC and other researchers. The authors should add a brief paragraph on this topic in the Discussion, to help the reader to better understand and evaluate various strategies.

Reviewer #3:

Remarks to the Author:

Chen and Gu and colleagues report on findings of PCR/NGS-based MRD of Ig rearrangements in a cohort of 399 children with B-ALL who underwent induction/consolidation following a unified pediatric ALL protocol. The novelty of the report was a descriptive evaluation of clonal IgK/IgL sequences in addition to IgH clonal sequences, including associating specific clonal types with outcomes. This was also done recently in a report of adult ALL patients by Liang and colleagues published in Blood Advances 2023. While this topic is very important, I felt the manuscript was confusing and in some areas lacked clarity, particularly around which group of patients was being evaluated in each particular section. Specific comments are below:

1. I do not see a definition for "low frequency clones". Typically, only dominant clones are tracked for MRD so I found these sections particularly confusing. The authors should very clearly define the methods of assigning dominant clone, low frequency clone, and what was actually tracked and considered MRD.

2. Similarly, I did not see any definition for limit of detection other than: "While these clonal sequences should occupy no less than 0.02% of total input Mononuclear 133 cells (determined by input DNA)." Was uniqueness of the sequence considered? Do all sequences have the same LOD? This would help as it is well known that IgH rearrangements are typically more unique than IgK/IgL sequences.

3. Some of these issues may be cleared up if the methods section was more clearly written. There are also tense changes and incomplete sentences that make reading this section difficult.

4. I had many questions and issues throughout the results section. In particular, I did not understand Figure 1B and how IgK/IgL allowed for tracking in an extra 5.5% of samples, given that it seems that in your cohort all patients had IgH clones +/- light chain clones in the text, which doesn't match Fig 1C.

5. In Figure 2, what do the red box plots represent? Perhaps add this to the figure legend as I could not figure it out.
6. Do your concordance plots include the "dominant" sequences and "low frequency" sequences?
7. Figure 4 and 5 should have number at risk tables
8. In the results section, "Prognostic Significance of IGH, IGK, and IGL MRD" what are you comparing in the IgK and IgL sentences? Is this positive and negative patients? It is entirely unclear as written and no Ns are provided unlike in the sentences for IgH.
9. The sankee diagrams in Fig 6 need better labeling to understand them.
10. The entire manuscript would benefit from additional editing as there are tense changes and sentence structure issues throughout. I would also change this "MRD < 0.0001%, 0.0001% ≤ MRD < 0.01%, and MRD ≥ 0.01%" nomenclature to MRD < 0.0001%, MRD ≥ 0.0001% and < 0.01% and MRD ≥ 0.01%

RESPONSE TO REVIEWERS

REVIEWER COMMENTS

Reviewer #1, expertise in pediatric B-ALL (Remarks to the Author):

1. The treatment protocol should have been in the supplementary material, but I was not able to find it. However, I understand from previous publications that B-ALL patients with MRD >1% at day 15 (TP1) or >0.1% at day 33 (TP2) of induction phase were allocated to the HR Group. Was NGS MRD used to stratify the patients of the cohort here reported? This aspect should be clarified in the manuscript.

Answer: Thank you very much for your comments. We are sorry for ignorance of uploading the supplementary material. The treatment protocol has been uploaded as supplementary material when revising the manuscript. Regarding the stratification of patients in the cohort reported here, NGS MRD was not used for patient stratification or treatment allocation. As you mentioned, we allocated the patients with MRD >1% at day 15 (TP1) or >0.1% at day 33 (TP2) of induction phase to the HR group. Here, the MRD referred to MRD detected by multiparameter flow cytometry or qPCR, but not NGS-MRD. The NGS-MRD has not been integrated into our protocol even now.

We added a supplementary file which contained a table to show how to allocated patients into LR, IR and HR group. In this table, we illustrated the MRD detection methods. And in methods section in line 92, we added: " In this protocol, NGS-MRD was not used for patient risk stratification or treatment allocation."

2. There is also an inconsistency on treatment; it is stated that patients in this cohort were treated according to the ZJCH-ALL-2019 protocol, while patients here reported were diagnosed since 2018.

Answer: We apologize for this confusion. The treatment protocol was initially implemented in our center since September 2018 and later extended to the whole Zhejiang Province in 2019. Therefore, we named this protocol as "ZJCH-ALL-2019" because it was used in multi-centers in 2019. This cohort included 6 newly diagnosed patients in November and December 2018.

To provide further clarity on this matter, we have added the following statement in the manuscript in line 90: "All enrolled patients were treated according to the ZJCH-ALL-2019 protocol detailed in the supplementary file. This protocol was implemented in our center since September 2018 and subsequently extended to the whole Zhejiang Province in 2019."

3. In respect of clinical outcomes and correlation with MRD findings it should be recognized that the follow-up time is rather short (median 20 months).

Answer: Thank you very much for your comments. We acknowledge this limitation and had a comment in the discussion part in line 452: "Our present study has certain limitations. First, the follow-up time of the cohort was relatively short. Although our result of the prognosis was consistent with the previous research, the exact conclusion needs to be observed for a longer period." In the revised manuscript, we add more

comments on this “Future studies with extended observation periods will provide a more comprehensive understanding of the prognostic significance of MRD monitoring in this patient population.”.

4. In the Discussion it is stated that in qPCR MRD monitoring “To reduce false-negative results arising from clonal evolution, two MRD markers with high sensitivity are generally required in clinical trials.” While this approach was used in the 2000, it is no longer applied; one sensitive marker is generally considered adequate for patients’ stratification.

Answer: Thank you for pointing out the change in MRD monitoring practices. We acknowledge that the approach is no longer mandatory but mentioned the sentence with respect to the scope of the current research in pediatric B-ALL.

The content in our manuscript (line 376) has been changed to “Ig and TCR gene rearrangements are clone-specific molecular markers used in MRD analysis. The use of oligoclonal Ig/TCR targets in MRD analysis may result in underestimation of the ALL load due to continuous and secondary rearrangements^{1,2}. Thus, two MRD markers with high sensitivity were recommended to reduce the false-negative results arising from clonal evolution in previous studies around or before the year 2000³. In contrast, NGS provides comprehensive analysis of all kinds of clonal rearrangements⁴. Whether the combination of IGH targets with other Ig rearrangements like IGK and IGL is a more suitable strategy for sensitive MRD detection in the era of NGS should be clarified”

5. Concerning the NGS MRD methodology used, there is no information about the presence of an internal standard, such as the spike-in of a DNA sample with known rearrangements of IGH/IGK/IGL and a known number of absolute copies of these rearrangements. The absence of such a type of spike-in, which instead is applied for example in the study by Svaton et al. (ref.16), may decrease the accuracy of results. Particularly at the EOI and EOC points, when the levels of disease are very low or negative, the amplification derived from MNC and residual blasts may not be linear, and this can limit the accuracy of MRD levels assessment. Thus, the authors should clarify if they used an internal calibrator and if not, whether they demonstrated the linearity of rearrangements amplification.

Answer: Thank you for your comments. We did not include known rearrangements spike-in DNA in each individual sample. However, we had internal standard in every batch of testing, by adding one positive control with known rearrangements spike-in DNA to assess the PCR amplification efficiency of the experiment. Additionally, during the development of the NGS MRD methodology, we conducted thorough optimization and validation processes.

To address potential primer amplification bias, we optimized the primers using over 200 known sequences (spike-in) synthesized from plasmids as standards. We made adjustments to primers with low amplification efficiency by selecting different PCR enzymes, buffers, temperatures, primers’ T_m, reaction cycle numbers and etc. Through the implementation of this optimization process, we successfully achieved impartial amplification. Also, during the actual sample testing process, this mixed known DNA served as a positive control to assess the amplification performance of each batch.

Furthermore, we evaluated the linearity of the NGS MRD method. We generated samples with known

numbers of absolute copies and rearrangements by mixing various cell line DNA samples with healthy individuals' PBMC DNA samples at different concentrations (200 ng, 2 µg, and 20 µg). Subsequently, we compared the detected results with the expected results. Remarkably, it was observed that the known rearrangements frequency exhibited linearity over several orders of magnitude under each testing condition, thus indicating excellent linearity in the detection capability of this method.

We have added the following content to the methods section in line 131.

“For enhanced result accuracy, each testing batch included an internal standard. This standard comprised a positive control with known rearrangements spike-in DNA, which allowed assessment of PCR amplification efficiency. Furthermore, during the development of the NGS MRD methodology, comprehensive optimization and validation processes were carried out to ensure reliability and validity. To mitigate primer amplification bias, primers were optimized using over 200 spike-in sequences synthesized from plasmids. Adjustments were made to primers with low amplification efficiency by varying PCR enzymes, buffers, temperatures, primer melting temperatures (T_m), and reaction cycle numbers. Mixed known DNA samples served as positive controls during sample testing to assess batch-specific amplification performance.

The linearity of the NGS MRD method was assessed by generating samples with known absolute copy numbers and rearrangements. Cell line DNA samples at different concentrations were mixed with healthy individuals' PBMC DNA samples. The detected results were compared to the expected results to confirm the linear relationship.”

6. It should be also clarified if data on MRD by PCR or FLOW are available in these patients, and if yes, to compare those data with NGS data to establish how they correlate with NGS findings.

Answer: Thank you very much for your advice. In order to let the readers understand the efficacy of NGS-MRD, it is very important to compare NGS to traditional MRD detection methods. Thus, we added these results following your advice in the revised manuscript in line 283. “In this cohort, MFC-MRD turned to be negative (<0.01%) in 95.2% of patients at EOI. MFC-MRD was prognostic at EOI as well, with patients with negative and positive MFC-MRD presenting 3-year EFS of $94.4\% \pm 1.8\%$ and $78.9\% \pm 11.4\%$ ($P = .008$) respectively (Figure 5A). However, NGS identified 93 patients whose MFC-MRD was negative while their NGS-MRD was >0.01%, which accounted for 26.2% of all patients with negative MFC-MRD. These patients presented much worse 3-year EFS than those with both MFC and NGS MRD negative ($84.7\% \pm 7.1\%$ vs. $97.0\% \pm 1.2\%$, $P = .008$). By contrast, there were only three patients whose NGS-MRD was negative but their MFC-MRD was >0.01%. All the three patients survived at the last follow up. Patients with both MFC and NGS MRD positive had the worst outcome, with 3-year EFS of $70.3 \pm 15.4\%$ only (Figure 5B).”

We added Figure 5A-5B in manuscript.

Figure 5

7. A novelty of this paper regards the findings on IGH vs. IGK/IGL, which suggest that IGK/IGL MRD findings are poorly correlated to prognosis, in favor of IGH. In the conclusion remarks the authors should indicate if in their opinion it is worth to continue pursuing IGK/IGL monitoring.

Answer: Thank you very much for your suggestion. We have added the following comment in the discussion part in the revised manuscript in line 460: “With high availability and sensitivity, the clonal rearrangement of IGH is the most important for MRD monitoring, whereas the clonal rearrangement of IGK and IGL was found only in approximately half of the cohort with very limited independent prognostic significance. Thus, we recommend continuing to pursue IGH monitoring but not IGK/IGL monitoring according to our experience.”

8. Editing suggestions:

Line 45, Background: “Monitoring the levels of minimal/measurable residual disease (MRD) defined as the presence of residual leukemic cells not detected by conventional tools, is a strong prognostic factor”: “Monitoring” should be “Detection”.

Line 93. “per the parent’s preference” should be “per parents’ preference”

Line 100: “The first-step PCR primers were aimed to acquire”: there is a need for editing; maybe “The first step was aimed to acquire PCR primers for”

Line 127: clonotypes should be clones

Line 154: “As a part of the initial visit,” should be “As a part of the initial diagnostics,”

Line 190: “The clearance of Ig clones was observed after chemotherapy.” maybe better “The clearance of Ig clones was assessed at sequential timepoints of chemotherapy.”

Line 191-192: both percentages should be referred to clones (and not to children)

Answer: Thank you for providing the editing suggestions. We have carefully reviewed the manuscript and the above-mentioned content has been corrected in the manuscript as suggested.

Reviewer #2, expertise in MRD testing and blood cancers (Remarks to the Author):

This is a large cohort of pediatric ALL patients evaluated with modern MRD assays. The design of the study is robust and the results are relevant. The paper is well written with balanced interpretations and references. The authors address the limitation that follow-up time of the cohort was relatively short.

Minor comments:

1. In the manuscript, the authors state "...A total of 430 pediatric patients with B-ALL were enrolled between November 2018 and April 2022. As a part of the initial visit, an Ig-/T-cell receptor (TCR) immune panel was analyzed to identify prognostic markers. Among these pediatric patients, 399 children (92.6%) had at least one Ig clonal rearrangement and were included for further analyses...". What is the reason for 7.4% of the cases not having at least one Ig clonal rearrangement? Sample quality or issues with the assay, or both? This should be briefly commented/clarified in the paper; both in the Results and Discussion sections.

Answer: Thank you for your question. The 7.4% of cases may have Ig clonal rearrangement, but did not have dominant clone for following tracking. We have criteria of dominant sequences following the FDA guideline: To define dominant clonal gene rearrangements in samples obtained at diagnosis, we used a frequency threshold of 3%. While these clonal sequences should occupy no less than 0.02% of total input mononuclear cells (determined by input DNA). And meanwhile, there was significant distance and no obvious regularity between the potential clonal sequences and the next several sequences subsequently. These criteria have been shown in the manuscript. There were 31 patients, although the Ig clonal rearrangements had been detected, but the clones did not fulfill the above criteria. Possible reasons for not detecting dominant Ig clonal rearrangement include the following: Some cases may not have detectable Ig clonal rearrangements because malignant cell transformation occurred prior to Ig/TCR gene rearrangements, rendering them undetectable through this method. Another possibility is that mutations occurred at the primer binding sites of the assay, resulting in ineffective and non-specific amplification.

We have revised our expression in the new manuscript to avoid confusion and discussed this in the Discussion section, the following explanation in line 385 has been provided: "In this cohort of pediatric patients with B-ALL, it was observed that 7.4% of the cases did not have dominant Ig clonal rearrangement for MRD tracking, which was similar to a previous report in adult ALL⁵. The possible reasons for this could be attributed to the following two aspects: First, malignant cells transformation may occur prior to Ig gene rearrangements, rendering them undetectable through this method^{6,7}. Secondly, mutations at the primer binding sites of the assay may result in ineffective and nonspecific amplification^{8,9}."

2. Would the authors be able to genomically profile cases with adverse (vs superior) clinical outcomes, e.g. using WGS? Similarly, can the authors profile host immune profiles in a similar manner? Such approaches would provide more detailed biological insights on biology in relation to outcomes. If possible, please add these data and expand the paper. If not possible, this should be commented briefly in the Discussion section.

Answer: Thank you for your suggestion. Following your suggestion, we have added a new figure in the revised manuscript. Herein, we compared the distributions of age, gender, initial WBC count, risk groups,

chromosome karyotype and fusion genes between patients with NGS-MRD $<0.01\%$ and $\geq 0.01\%$ at EOI, and between patients with NGS-MRD $<0.0001\%$ and $\geq 0.0001\%$ at EOC.

We have added the following content to the result section in line 256.

“We first investigated the associations between clinical/laboratory features and NGS-MRD eradication. As shown in Figure 4A and 4B, patients aged older than 10 years or assigned into higher-risk group presented higher NGS-MRD levels at EOI, and patients in the high-risk group still had higher NGS-MRD levels at EOC. Regarding the genetic features, patients with *ETV6/RUNX1* presented much lower NGS-MRD levels than other fusion genes including *BCR/ABL1*, *TCF3/PBX1* and *MLL*-rearrangements (Figure 4C), and patients with *MLL*-rearrangements were more likely to have higher NGS-MRD levels compared to other fusion genes, including *ETV6/RUNX1*, *BCR/ABL1*, and *TCF3/PBX1* at EOC (Figure 4D). The chromosome karyotype seemed not to be associated with NGS-MRD level at both EOI and EOC (Figure 4C and 4D).”

Figure 4. The relationship between NGS-MRD and baseline clinical/laboratory features and survival. Distributions of gender, age, risk groups, initial WBC count, fusion genes and chromosome karyotype between patients with NGS-MRD $<0.01\%$ and $\geq 0.01\%$ at EOI (A, C), and between patients with NGS-MRD $<0.0001\%$ and $\geq 0.0001\%$ at EOC (B, D) were compared. Asterisks indicate statistical significance at $P < 0.05$, determined using the chi-square or Fisher's exact test. The P-values obtained from comparing the composition ratios of all subgroups within this category.

3. How do the applied MRD assays perform in relation of other available MRD assays capturing IGH and K/L, such as Adaptive Clonoseq, Invivoscribe, etc? How are the assays similar/different? What data is available from ALL patients (or other B-cell malignancies) using these type of assay? For example, work has been done in lymphomas by MSKCC, Stanford, and NIH/NCI researchers, and work has been done in myeloma by MSKCC and other researchers. The authors should add a brief paragraph on topic in the Discussion, to help the reader to better understand and evaluate various strategies.

Answer: In addition to our Seq MRD testing method, there are other available MRD assays that capture the immunoglobulin heavy chain (IGH) and kappa/lambda light chain (K/L) loci, such as Adaptive Clonoseq and

Invivoscribe. These three methods share a similar overall objective of detecting and monitoring minimal residual disease (MRD) in B-cell malignancies, and they all demonstrate high sensitivity reaching 10^{-6} .

Seq-MRD and Clonoseq share similarities in their detection processes. Both methods employ consensus primers targeting the IGH (VDJ), IGH(DJ), IGK, and IGL loci. They utilize sequencing data to identify clonotypes and quantify their frequencies. Seq-MRD includes additional targets such as IGKDE, while Clonoseq includes translocated BCL1/IgH(J) and BCL2/IgH(J) as well. On the other hand, Invivoscribe's testing is more limited, targeting only the IGH and IGK loci. The method uses commercial primer sets and is designed to amplify V(D)J rearrangements. Moreover, the definition of dominant clone for following tracking was the same between Seq-MRD and Clonoseq, which is different from that of Invivoscribe^{10, 11}.

Regarding the available data, research studies have investigated the performance and prognostic value of these MRD assays in various B-cell malignancies. For example, a study specifically focused on diffuse large B-cell lymphoma (DLBCL) to evaluate the role of MRD in predicting relapse and guiding treatment decisions¹². Additionally, MSKCC has conducted research in multiple myeloma, with a particular emphasis on the detection and tracking of MRD using Clonoseq and other methods^{13, 14}.

However, it should be noted that the specific comparison of Seq-MRD with other assays, such as Adaptive Clonoseq or Invivoscribe, including their performance and concordance, has not been extensively reported in the available literature. Additionally, the export of clinical research trial genetic data from China is subject to strict regulations and Adaptive Clonoseq or Invivoscribe has not been introduced into China, thus limiting the availability of data comparing Seq-MRD with these methods in China.

We have added the following content to the discussion section in line 436.

“In addition to Seq-MRD testing method used in this study, there are other available MRD assays that capture the immunoglobulin heavy chain (IGH) and kappa/lambda light chain (K/L) loci, such as Adaptive Clonoseq and Invivoscribe. These three methods share a similar overall objective of detecting and monitoring MRD in B-cell malignancies, and they all demonstrate high sensitivity reaching 10^{-6} . Seq-MRD and Clonoseq target the IGH (VDJ), IGH(DJ), IGK, and IGL loci to identify clonotypes using sequencing data, while Invivoscribe focuses on the IGH and IGK loci. Moreover, the definition of the dominant clone for following tracking was the same between Seq-MRD and Clonoseq, which is different from that of Invivoscribe^{10, 11}

NGS for MRD assessment is applicable to various B-lineage malignancies except for ALL, such as chronic lymphocytic leukemia (CLL), multiple myeloma (MM) and lymphoma^{9, 12, 15, 16}. It plays a crucial role in improving risk stratification, treatment response monitoring, and prognostic outcomes^{16, 17}. NGS enables more sensitive and specific detection of MRD, allowing for precise risk assessment and identification of patients at higher risk of relapse^{18, 19}. It facilitates personalized treatment strategies, such as targeted therapies and transplantation, based on MRD status^{12, 20}. NGS-based MRD assessment provides valuable information for clinical decision-making and enhances the management of these diseases”.

Reviewer #3, expertise in MRD and blood cancers (Remarks to the Author):

Chen and Gu and colleagues report on findings of PCR/NGS-based MRD of Ig rearrangements in a cohort of 399 children with B-ALL who underwent induction/consolidation following a unified pediatric ALL protocol. The novelty of the report was a descriptive evaluation of clonal IgK/IgL sequences in addition to IgH clonal sequences, including associating specific clonal types with outcomes. This was also done recently in a report of adult ALL patients by Liang and colleagues published in Blood Advances 2023. While this topic is very important, I felt the manuscript was confusing and in some areas lacked clarity, particularly around which group of patients was being evaluated in each particular section. Specific comments are below:

1. I do not see a definition for "low frequency clones". Typically, only dominant clones are tracked for MRD so I found these sections particularly confusing. The authors should very clearly define the methods of assigning dominant clone, low frequency clone, and what was actually tracked and considered MRD.

Answer: Thank you very much for your comment. We are sorry for the confusion. In order to investigate the association between clearance rapidity and clone frequency, we defined the clones with frequency less than 10% as low frequency clone, and compared the clearance time with the highest-frequency clones. In Figure 2G,2I and 2K, the dominant clones referred to the clones with highest frequencies in each patient. Indeed, this is confused with another dominant clone define as frequency >3%. So, in the revised manuscript, we compared the clearance time of clones with frequencies <10% to that of clones with frequencies >=50%. We hope it will be not so confusing after revising.

2. Similarly, I did not see any definition for limit of detection other than: "While these clonal sequences should occupy no less than 0.02% of total input Mononuclear 133 cells (determined by input DNA)." Was uniqueness of the sequence considered? Do all sequences have the same LOD? This would help as it is well known that IgH rearrangements are typically more unique than IgK/IgL sequences.

Answer: In this manuscript, we would like to clarify that the term "limit of detection" mentioned in line 349 in previous manuscript should be interpreted as "cutoff". In our study, we evaluated EOC-MRD using a stringent cutoff. At the EOI, children with MRD cutoff <0.01% have a favorable prognosis. However, at the EOC time point, the MRD cutoff must be <0.0001%. We have deleted this "limit of detection" in the revised manuscript to avoid confusion.

We also took into consideration the uniqueness of all dominant clones. We have established a healthy donor's Ig database. To assess the probability of the detected dominate clone sequences (>3%) occurring in a healthy donor's Ig database we established, we compared them and assigned a uniqueness index to each dominate clone sequence. This uniqueness index helps evaluate the likelihood of the sequence appearing in the non-ALL repertoire. Clones with low uniqueness indices, suggesting potential associations with other diseases among non-ALL individuals, are excluded from MRD tracking. This approach helps eliminate some false cancer cell clones.

Different clone sequences in the tested samples have varying LoD. When evaluating the LoD for trackable sequences, we considered several factors, including the amount of DNA input, sequence uniqueness, as well as the diversity and mutation of those sequences.

We have added the following content to the methods section in line 144.

“The uniqueness of the dominant clone sequences was assessed by establishing a healthy donor Ig database. The probability of occurrence of the detected dominant clone in the healthy donor Ig database was investigated, and each dominant clone sequence was assigned a uniqueness index. This index provided an evaluation of the likelihood of the sequence being present in the non-ALL repertoire. Clones with low uniqueness indices were excluded from MRD tracking. Different clone sequences in the tested samples have varying limit of detection (LoD). When determining the LoD for trackable sequences, several factors were taken into account, including the amount of DNA input, sequence uniqueness, as well as the diversity and mutation of those sequences.”

3. Some of these issues may be cleared up if the methods section was more clearly written. There are also tense changes and incomplete sentences that make reading this section difficult.

Answer: Thank you for your comments. We have revised the methods part carefully. We hope it will be more clear after revising.

4. I had many questions and issues throughout the results section. In particular, I did not understand Figure 1B and how IgK/IgL allowed for tracking in an extra 5.5% of samples, given that it seems that in your cohort all patients had IgH clones +/- light chain clones in the text, which doesn't match Fig 1C.

Answer: Based on Figure 1C, it is evident that 22 patients (1 IGL only, 2 IGK and IGL, 19 IGK only) did not have IgH clones. If only IgH clones were detected, these 22 out of 399 patients (5.5%) would not have dominant Ig clones for tracking. Therefore, the inclusion of IGK/IGL rearrangements allowed for the detection of additional clones in these 5.5% of patients.

However, there may some confusion in the expression, thus we change this sentence in line 194 to “Twenty-two patients did not have trackable IgH clone but had IGL and/or IGK for tracking. Thus, the IGK/IGL rearrangements allowed tracking an additional 5.5% of samples that would have been uninformative if only the IGH assay was used for MRD monitoring.”

5. In Figure 2, what do the red box plots represent? Perhaps add this to the figure legend as I could not figure it out.

Answer: We apologize for the confusion. The red box plots in Figure 2 represent the distribution of data within the interquartile range (IQR) for the respective groups or categories being compared. The box itself represents the IQR, which includes the middle 50% of the data. The line or asterisk within the box indicates the median value. Additionally, the lines extending from the box, known as whiskers, represent the minimum and maximum values within a certain range, which is determined as 1.5 times the IQR. Any data points outside this range are considered outliers.

To clarify this in the figure legend, we have added the following information in figure 2 legend: " Dotted lines connect the clones of the same patient at different timepoints. The box plots represent the interquartile range (IQR) of the data. The line and asterisk within the box indicate the median value. The lines extending from the box, known as whiskers, represent the 1.5 times the IQR. Any data points

outside this range are considered outliers.”

6. Do your concordance plots include the "dominant" sequences and "low frequency" sequences?

Answer: Yes, the concordance plots in Figure 3 include both the "dominant" sequences and the "low frequency" sequences. All clones with frequency higher than 3% and fulfilling the criteria of trackable clone were included for analyzing.

7. Figure 4 and 5 should have number at risk tables

Answer: Thank you for your suggestion. Number at risk tables have been added to Figure 4 and 5.

8. In the results section, "Prognostic Significance of IGH, IGK, and IGL MRD" what are you comparing in the IgK and IgL sentences? Is this positive and negative patient? It is entirely unclear as written and no Ns are provided unlike in the sentences for IgH.

Answer: In the results section, "Prognostic Significance of IGH, IGK, and IGL MRD," the IgK and IgL sentences compare different patient groups based on their MRD levels. Specifically, we compare the prognostic significance of IgK and IgL levels in relation to clinical outcomes. The IgK and IgL patient groups are categorized based on their MRD status, including MRD $\geq 0.0001\%$ and $< 0.01\%$, MRD $< 0.01\%$ and MRD $\geq 0.01\%$. We apologize for any previous ambiguity. To provide clarity, we have included the sample sizes (n) in figure 6 in the revised manuscript to better describe the patient cohorts in each group.

9. The sankey diagrams in Fig 6 need better labeling to understand them.

Answer: Thank you for your feedback. The Sankey diagrams in Figure 6 (now is Figure 7) have been improved with better labeling to enhance understanding.

10. The entire manuscript would benefit from additional editing as there are tense changes and sentence structure issues throughout. I would also change this "MRD $< 0.0001\%$, $0.0001\% \leq \text{MRD} < 0.01\%$, and MRD $\geq 0.01\%$ " nomenclature to MRD $< 0.0001\%$, MRD $\geq 0.0001\%$ and $< 0.01\%$ and MRD $\geq 0.01\%$

Answer: Thank you for your valuable feedback. We have carefully reviewed and edited the manuscript to address the tense changes and sentence structure issues. Additionally, we have made the necessary modifications to the MRD nomenclature as suggested, changing it to "MRD $< 0.0001\%$, MRD $\geq 0.0001\%$ and $< 0.01\%$, and MRD $\geq 0.01\%$ ".

References

1. Gawad C, *et al.* Massive evolution of the immunoglobulin heavy chain locus in children with B precursor acute lymphoblastic leukemia. *Blood* **120**, 4407-4417 (2012).
2. Wu D, *et al.* Detection of minimal residual disease in B lymphoblastic leukemia by high-throughput

- sequencing of IGH. *Clin Cancer Res* **20**, 4540-4548 (2014).
3. Szczepanski T, Flohr T, van der Velden VH, Bartram CR, van Dongen JJ. Molecular monitoring of residual disease using antigen receptor genes in childhood acute lymphoblastic leukaemia. *Best Pract Res Clin Haematol* **15**, 37-57 (2002).
 4. Brüggemann M, *et al.* Standardized next-generation sequencing of immunoglobulin and T-cell receptor gene recombinations for MRD marker identification in acute lymphoblastic leukaemia; a EuroClonality-NGS validation study. *Leukemia* **33**, 2241-2253 (2019).
 5. Liang EC, *et al.* Next-Generation Sequencing-Based MRD in Adults with ALL Undergoing Hematopoietic Cell Transplantation. *Blood Adv*, (2023).
 6. Wood B, *et al.* Measurable residual disease detection by high-throughput sequencing improves risk stratification for pediatric B-ALL. *Blood* **131**, 1350-1359 (2018).
 7. Hussaini MO, *et al.* Assessment of Clonotypic Rearrangements and Minimal Residual Disease in Lymphoid Malignancies. *Arch Pathol Lab Med* **146**, 485-493 (2022).
 8. Rustad EH, Boyle EM. Monitoring minimal residual disease in the bone marrow using next generation sequencing. *Best Pract Res Clin Haematol* **33**, 101149 (2020).
 9. Medina A, *et al.* Comparison of next-generation sequencing (NGS) and next-generation flow (NGF) for minimal residual disease (MRD) assessment in multiple myeloma. *Blood Cancer J* **10**, 108 (2020).
 10. Arcila ME, *et al.* Establishment of Immunoglobulin Heavy (IGH) Chain Clonality Testing by Next-Generation Sequencing for Routine Characterization of B-Cell and Plasma Cell Neoplasms. *J Mol Diagn* **21**, 330-342 (2019).
 11. Monter A, Nomdedéu JF. ClonoSEQ assay for the detection of lymphoid malignancies. *Expert Rev Mol Diagn* **19**, 571-578 (2019).
 12. Merryman RW, *et al.* Minimal residual disease in patients with diffuse large B-cell lymphoma undergoing autologous stem cell transplantation. *Blood Adv*, (2022).
 13. Hulcrantz M, *et al.* Capture Rate of V(D)J Sequencing for Minimal Residual Disease Detection in Multiple Myeloma. *Clin Cancer Res* **28**, 2160-2166 (2022).
 14. Ho C, *et al.* Routine Evaluation of Minimal Residual Disease in Myeloma Using Next-Generation Sequencing Clonality Testing: Feasibility, Challenges, and Direct Comparison with High-Sensitivity Flow Cytometry. *J Mol Diagn* **23**, 181-199 (2021).
 15. Hengeveld PJ, *et al.* Detecting measurable residual disease beyond 10⁻⁴ by an IGHV leader-based NGS approach improves prognostic stratification in CLL. *Blood* **141**, 519-528 (2023).
 16. Svaton M, *et al.* NGS better discriminates true MRD positivity for the risk stratification of childhood ALL treated on an MRD-based protocol. *Blood* **141**, 529-533 (2023).
 17. Li Z, *et al.* Identifying IGH disease clones for MRD monitoring in childhood B-cell acute lymphoblastic leukemia using RNA-Seq. *Leukemia* **34**, 2418-2429 (2020).
 18. Anderson KC, *et al.* The Role of Minimal Residual Disease Testing in Myeloma Treatment Selection and Drug Development: Current Value and Future Applications. *Clin Cancer Res* **23**, 3980-3993 (2017).
 19. Short NJ, *et al.* High-sensitivity next-generation sequencing MRD assessment in ALL identifies patients at very low risk of relapse. *Blood Adv* **6**, 4006-4014 (2022).

20. Fonseca R, *et al.* Integrated analysis of next generation sequencing minimal residual disease (MRD) and PET scan in transplant eligible myeloma patients. *Blood Cancer J* **13**, 32 (2023).

Reviewers' Comments:

Reviewer #1:

Remarks to the Author:

Replies to my requests are adequate. I suggest, however, to include the experimental details supporting the quality of the data in the supplementary material section and to add in the text a comment on the stability of IGK/IGL markers.

Reviewer #2:

Remarks to the Author:

None

Reviewer #3:

Remarks to the Author:

The authors did quite a bit of work to answer the reviewers questions and comments, which is appreciated. Very small point, but I do not think the term "nearly curable" disease in the abstract is appropriate as a) I do not know what that means and b) the fup time is short so we really do not know who may be "cured".

Reviewer #1 (Remarks to the Author):

Replies to my requests are adequate. I suggest, however, to include the experimental details supporting the quality of the data in the supplementary material section and to add in the text a comment on the stability of IGK/IGL markers.

Answer: Thank you for your suggestion.

Regarding quality of the data, we conducted a comprehensive analysis on 399 initial diagnosis (Pre) samples and 734 follow-up (Post) samples, focusing on the following aspects:

a. **Number of Input Cells for Sequencing:** The number of input cells was calculated based on the amount of DNA used in each experiment, standardized at 6.4 pg per cell. For the Pre samples, the primary objective was to identify dominant clones. To that end, we opted for a smaller sample size dictated by the sampling situation. In contrast, for the Post samples, our aim was to maximize the sensitivity of our detection methods; therefore, we used as many samples as clinically feasible.

b. **Size of Sequencing Raw Data:** This metric refers to the number of FASTQ reads generated for each sample. Due to our aim of maximizing sensitivity in the Post samples, the raw FASTQ data size for these samples was larger than that for the Pre samples.

c. **Q30 Value of the FASTQ Data:** The Q30 value serves as a standard measure for evaluating the quality of the base calls during sequencing. The specific information can be found in Table 1.

Regarding stability of IGK/IGL markers, we independently analyzed data from both initial diagnosis (Pre) and follow-up (Post) samples to determine the clonal counts of IGK and IGL. Due to variations in the treatments administered to different patients, quantitative discrepancies may manifest in the raw data. For specific results, please refer to Table 1.

Table 1. Characteristics of Pre and Post sample.

Index	Pre	Post
Average total cell counts	272908 ± 166311	1361670 ± 1075839
Quality of the data		
Average raw data read counts(G)	1.76 ± 1.54	3 ± 3.55
Average raw data Q30(%)	92.49 ± 2.68	92.7 ± 2.86
Stability of		
Average IGK clone counts	1385373 ± 2012618	1406895 ± 1722960
IGK/IGL markers Average IGL clone counts	1062789 ± 2136410	1089539 ± 1477219

To statistically assess the data quality and stability, the data related to total cell counts, raw data read counts, raw data Q30 values, IGK counts, and IGL counts underwent quality assessment utilizing the 3-sigma (3σ) statistical method^{1, 2, 3}. It's noteworthy that the 3σ principle is based on a normal distribution, where roughly 68% of observations fall within one standard deviation (σ) from the mean, about 95% within

two standard deviations, and approximately 99.7% within three standard deviations. In practical applications, especially with limited sample sizes, deviations may occur. Initially, we performed a normality test on five data sets. As evidenced by Table 2, all five sets demonstrated substantial normality with a P -value less than 0.0001, indicating a significant normal distribution for each set.

Table 2. Normality test results of five metrics.

	Time Point	Shapiro-Wilk		
		Statistics	df	Sig.
Total cell counts	Pre	.557	396	.000
	Post	.910	693	.000
Raw data read counts	Pre	.758	362	.000
	Post	.519	684	.000
Raw data Q30	Pre	.950	395	.000
	Post	.911	724	.000
IGK clone counts	Pre	.761	570	.000
	Post	.713	306	.000
IGL clone counts	Pre	.461	357	.000
	Post	.695	629	.000

Subsequently, we conducted a more in-depth examination of the standard deviations using the 3-sigma ($3\text{-}\sigma$) test principle, as depicted in Figure 1. In both Pre and Post samples, more than 95% of the data fell within the $\mu\pm 2\sigma$ range, and roughly 99% within the $\mu\pm 3\sigma$ range. Although there was a minor deviation from the theoretical 99.7% within the $\mu\pm 3\sigma$ range, attributed to natural fluctuations in the data, the results were still in alignment with the normality test.

In summary, the analysis confirms that our experimental methods are consistent, and consequently, the quality of our data is robust and reliable. Moreover, the IGK and IGL data demonstrated good stability across different patients and time points, reinforcing the reliability of our findings.

Figure 1. Illustrations of the comprehensive data quality assessment. Data were performed on five metrics: Total Cell Counts (A), Raw Data Read Counts (B), Raw Data Q30 Scores (C), IGH Clone Counts (D), and IGL Clone Counts (E), using the 3-Sigma ($3\text{-}\sigma$) statistical test principle. The x-axis differentiates between Pre and Post sample groups, with individual samples denoted by black data points. The y-axis quantifies the Z-score, computed as $Z = (X - \mu) / \sigma$. Various threshold lines are color-coded: the blue dashed line marks the $\mu \pm 1\sigma$ range, the green dashed line indicates the $\mu \pm 2\sigma$ range, the orange dashed line signifies the $\mu \pm 3\sigma$ range, and the red dashed line delineates the position of the average standard deviation, denoted by μ .

All the above information has been put in the supplementary file. Furthermore, we have added the following content to the results section.

“Data Quality Evaluation: To statistically assess the data quality and stability across different time points and patient samples, the data for (1) number of input cells for sequencing, (2) size of sequencing raw data, (3) Q30 value of the FASTQ data, (4) IGH counts, and (5) IGL counts were subjected to data quality assessment using a 3-sigma ($3\text{-}\sigma$) test principle (Supplementary Tables 1, 2 and Supplementary Figure 1). The results confirmed the stability and reliability of the data. The IGH and IGL data from different patients at both diagnosis and follow up time points demonstrate good stability as well.”

Reviewer #2 (Remarks to the Author):

None

Reviewer #3 (Remarks to the Author):

The authors did quite a bit of work to answer the reviewers questions and comments, which is appreciated. Very small point, but I do not think the term "nearly curable" disease in the abstract is appropriate as a) I do not know what that means and b) the fup time is short so we really do not know who may be "cured".

Answer: Thank you for your suggestion. We have revised the term in the abstract to "Patients with NGS-MRD <0.01% at EOI or <0.0001% at EOC present excellent outcome, with 3-year event-free survival rates higher than 95%".

References

1. Nyhuis P, Wiendahl H-P. 3-Sigma PPC - A Holistic Approach for Managing the Logistic Performance of Production Systems. *CIRP Annals* **53**, 371-376 (2004).
2. Shannaq B, Al-Azzawi F. Three-Sigma Scale Model for Measuring Student Interest in Social Media-Effective Tool for Improving the Educational Process in the Coronavirus (Covid-19) Period. *International Journal of Advanced Science and Technology* **29**, 3597-13609 (2020).
3. Xiao H, Zhang Y, Liu X, Yin H, Liu P, Liu DC. A Rapid Ultrasound Vascular Disease Screening Method using PauTa Criterion. *Journal of Physics: Conference Series* **1769**, (2021).